



# Geometric estimation of volcanic eruption column height from GOES-R near-limb imagery – Part 2: Case studies

Ákos Horváth[1], Olga A. Girina[2], James L. Carr[3], Dong L. Wu[4], Alexey A. Bril[5], Alexey A. Mazurov[5], Dmitry V. Melnikov[2], Gholam Ali Hoshyaripour[6], Stefan A. Buehler[1]

[1]Meteorological Institute, Universität Hamburg, Hamburg, Germany
[2]Institute of Volcanology and Seismology, Far East Branch of the Russian Academy of Sciences (IVS FEB RAS), Petropavlovsk-Kamchatsky, Russia
[3]Carr Astronautics, Greenbelt, MD, USA
[4]NASA Goddard Space Flight Center, Greenbelt, MD, USA
[5]Space Research Institute of the Russian Academy of Sciences (SRI RAS), Moscow, Russia
[6]Institute of Meteorology and Climate Research, Karlsruhe Institute of Technology (KIT), Karlsruhe, Germany

*Correspondence to*: Ákos Horváth (akos.horvath@uni-hamburg.de, hfakos@gmail.com)

**Abstract.** In a companion paper (Horváth et al., 2021), we introduced a new technique to estimate volcanic eruption column height from extremely oblique near-limb geostationary views. The current paper demonstrates and validates the technique in a number of recent eruptions, ranging from ones with weak columnar plumes to subplinian events with massive umbrella clouds and overshooting tops that penetrate the stratosphere. Due to its purely geometric nature, the new method is shown to be unaffected by the limitations of the traditional brightness temperature method, such as height underestimation in subpixel and semitransparent plumes, ambiguous solutions near the tropopause temperature inversion, or the lack of solutions in undercooled plumes. The side view height estimates were in good agreement with plume heights derived from ground-based video and satellite stereo observations, suggesting they can be a useful complementary to established techniques.

## 1 Introduction

In Part 1, we gave a detailed description of our geometric method that exploits the near-limb portion of daytime geostationary imagery to derive point estimates of eruption column height in the vicinity of the vent. Such oblique observations offer close to orthogonal side views of vertical columns protruding from the Earth ellipsoid and thereby facilitate a simple height-by-angle technique. In principle, the method can be applied to data from any geostationary instrument, but in practice it is best suited to the latest generation Advanced Baseline Imager (ABI) onboard the Geostationary Operational Environmental Satellite-R (GOES-R) series and the almost identical Advanced Himawari Imager (AHI) onboard the Himawari 3rd generation satellites. The ABI and AHI provide currently the finest visible channel geostationary imagery (500 m resolution at the subsatellite point) and, in the case of the former, best-in-class georegistration. Part 1 also included a preliminary validation by using the proposed method to determine the height of known Kamchatkan mountain peaks identified in GOES-17 images.



In Part 2, we apply the side view height estimation to seven volcanic eruptions from 2019 and 2020, which were observed by GOES-17 near the limb of the full disk image. Our goal is to further test the method and demonstrate its strengths and limitations in a variety of cases that represent a range of explosivity, plume morphology, and observing conditions. The side view plume heights are compared with heights from the basic temperature method, GOES-17–Himawari-8 stereo retrievals, and where available estimates from ground-based video footage.

## 2 Notes on VolSatView and the GOES-17 loop heat pipe anomaly

Two technical points are worth mentioning about the temperature methods used for comparison. First, the VolSatView information system (Bril et al., 2019; Girina et al., 2018; Gordeev et al. 2016) operated by the Kamchatka Volcanic Eruption Response Team (KVERT) derives plume heights from Himawari-8 11 μm brightness temperatures ($BT_{11}$). The algorithm uses forecast profiles from the Historical Unidata Internet Data Distribution (IDD) Gridded Model Data archive, given at 1° spatial and 3-hour temporal resolution. A temperature profile is interpolated to the geographic location and acquisition time of a pixel and then plume height is set to the interpolated profile altitude whose temperature matches the measured $BT_{11}$. A standard atmosphere profile can alternatively be selected for the retrievals, a choice often exercised to mitigate height underestimation in small thin plumes.

Second, we also estimate plume height by matching GOES-17 dark pixel $BT_{11}$ to ERA5 temperature profiles. The GOES-17 ABI unfortunately suffered a loop heat pipe (LHP) anomaly, which prevented cooling the thermal focal plane to its operational temperature. After a number of performance recovery steps, 97% of imaging capability in the thermal infrared bands was regained (McCorkel et al., 2019). To minimize any potential effects of the LHP anomaly on our results, we only use ABI brightness temperatures with a data quality flag (DQF) of 0, indicating 'good pixels'.

## 3. Case studies

We analyze eruption plumes from Sheveluch, Bezymianny, and Karymsky in Kamchatka, Raikoke in the Kurils, and Ulawun in Papua New Guinea, all located near the limb of the GOES-17 full disk image. The GOES-17 and Himawari-8 view geometries for these volcanoes are plotted in Fig. 1. GOES-17 observes the volcanoes in Kamchatka and the Kurils from east-southeast at very oblique view zenith angles (VZAs) of 83º–86º, while Himawari-8 observes them from south-southwest at much smaller VZAs of 57º–67º. Ulawun, situated slightly south of the equator, is viewed almost exactly from the east at a VZA of 80º by GOES-17 and from the northwest at a near-nadir VZA of 14º by Himawari-8.

### 3.1 Sheveluch, 8 April 2020

As briefly discussed in Part 1, an explosive eruption of the volcano occurred on 8 April 2020 sometime between 19:00 UTC and 19:10 UTC, slightly after sunrise (solar zenith angle ~84º). The Kamchatka Volcanic Eruption Response Team



(KVERT) issued an orange coded Volcano Observatory Notice for Aviation (VONA 2020-40, http://www.kscnet.ru/ivs/kvert/van/?n=2020-40), reporting a plume height of 9.5–10.0 km as determined by the basic temperature method from Himawari-8 11 µm data. This eruption is well suited to a detailed analysis, because it produced a single puff in a weak wind field, resulting in a nearly vertical eruption column with minimal bending and an isotropically-

spread umbrella cloud (strong plume). The GOES-17 visible band fixed grid images covering the eruption and the initial evolution of the plume between 19:10–19:30 UTC are plotted in Fig. 2a-c, while the estimated eruption geometry is sketched in Fig. 2d. The advection and dispersion of the plume in the first five hours after the eruption is depicted in Supplementary Animation 1, which shows the GOES-17 and Himawari-8 images in equirectangular map projection as well as the GOES-17 fixed grid view—degassing and a weak plume from Klyuchevskoy, especially after 23:00 UTC, is also evident. The

Himawari-8 fixed grid view is not dramatically different from the equirectangular map thanks to the smaller view zenith angles and, hence, is excluded.

As shown, the plume was advected south-southeast by the wind and eventually drifted over the ocean. The gradual tilting of the plume and expansion of the umbrella cloud are apparent in the 19:20–19:30 UTC fixed grid images, which also hint at faint ash fallstreaks underneath the southern half of the umbrella cloud. The animation vividly demonstrates the vastly

different and complementary perspectives offered by GOES-17 and Himawari-8: the former observes the plume from the east-southeast (~114º azimuth) at very oblique view zenith angle (VZA, ~84º), while the latter provides a view from the south-southwest (about -156º azimuth) at much smaller but still considerable VZA (~67º). For GOES-17 near-limb data the fixed grid image is a more natural and easier-to-interpret representation than equirectangular remapping, which suffers from extreme zonal stretching. A notable consequence of the different view geometries is that the dispersing plume becomes semi-

transparent in Himawari-8 images by the time it reaches the ocean, but it remains opaque in GOES-17 images due to the longer slant path. The GOES-17 fixed grid side view clearly shows the advecting ash volume splitting into two vertically distinct layers due to wind shear: a faster moving upper layer of the umbrella cloud and a slower moving lower layer that developed from the 'stem' of the initial mushroom cloud. The superior georegistration of GOES-17 compared to Himawari-8 is also apparent in the time-lapse imagery.

The geometry of the plume at the time of the eruption is sketched in Fig. 2d. As shown in Fig. 10d of Part 1 and also in Fig. 3 of the current paper, the side view method yields a plume height of ~8 km above the ellipsoid and, thus, a column height of ~5.5 km above the vent. The umbrella cloud can be reasonably well fitted with a circle of ~5.9 km radius, which appears distorted as a large eccentricity ellipse near the limb of the fixed grid projection. The geographical coordinates of the fitted finite circle are calculated from the direct solution on the sphere as described by Bildirici and Ulugtekin (2011) and

Bildirici (2015). Such geometry estimates offered by the oblique side view might be useful in the evaluation of plume simulations. For example, the radius to height aspect ratio of ~1.07 determined for this plume is in good agreement with the ~1.06 near-field aspect ratio obtained by Burton et al. (2020) in their idealized simulations of a buoyant gas plume in a quiescent atmosphere. The ERA5 profiles show north-northwesterly winds that increase from 4 m s$^{-1}$ to 9 m s$^{-1}$ between the vent and the plume top. The average plume rise speed is ~9.2 m s$^{-1}$, although this likely is an underestimate because the



plume reached its maximum altitude in less than 10 minutes, as evidenced by the spread of the umbrella cloud. Compared to the layer-mean wind of ~6.5 m s$^{-1}$ this rise speed indicates a strong eruption; as a result, the center of the plume was shifted south of the vent by only ~2.5 km at 19:10 UTC.

     The side view method requires the plume top position right above the vent. Our visual estimate of this point is marked by the blue diamond in Fig. 2d, which lies halfway between the far-side (western) and near-side (eastern) edges of the umbrella

cloud. When the isoheights are drawn relative to the volcano base (default), this plume location yields a height estimate of ~8 km, as shown in Fig. 3b. The far-side and near-side plume edges, however, appear at ~9 km and ~7 km, respectively, due to retrieval biases caused by the radial expansion as explained in section 4.1 of Part 1. The latitudinal (south-north or left-right) expansion/advection of the plume, in contrast, has little effect on the height estimates. The far-side/near-side plume edge is located behind/in front of the plane of isoheights drawn in Fig. 3b. These height biases due to the westward/eastward

plume expansion can be roughly corrected for by shifting the isoheights plane accordingly, as demonstrated in Fig. 3a and Fig. 3c. Shifting the baseline (zero line) to the westernmost (Fig. 3a) or easternmost (Fig. 3c) point of the surface projection of the circular plume top fit (white dashed line), leads to height estimates of ~8 km for the edges of the umbrella cloud too.

     The side view height estimate can be compared with estimates from traditional geometric methods (see section 3 of Part 1), which are listed for the current and all subsequent eruptions in Table 1. The GOES-17 and Himawari-8 visible band

images at 19:10 UTC are plotted in equirectangular map projection in Fig. 4. The location of characteristic features of the plume and its shadow, as well as the vectors between these points, are also indicated. Compared to Himawari-8, the GOES-17 image of the low-level cloud layer is rather blurred and shows little detail due to the large near-limb GIFOV (ground-projected instantaneous field of view). More detail and texture are seen in the image of the vertical eruption column and the mountains thanks to the fine VIFOV (vertically-projected instantaneous field of view), but the equirectangular remapping

generally suffers from extreme zonal stretching near the limb. This zonal stretching also leads to non-negligible azimuth distortions, which have to be considered when measuring distances and directions between points (see section 3.5 in Part 1). For example, there is a 10°/15° difference between the true and mapped view azimuths for GOES-17/Himawari-8. The true solar azimuth is close to east and thus has small (1°–2°) distortion.

     Height can be estimated from the sensor-projected column length measured along the map distorted view azimuth

(method 1). The good alignment of the map projection of the eruption column with the distorted view azimuth direction indicates little tilting and advection relative to the vent. The ellipsoid distance between the volcano base (red triangle) and the plume center (or center of the fitted circle, blue dot) is respectively ~75.7 km and ~18.8 km for GOES-17 and Himawari-8. Here the distance to the center of the umbrella cloud is needed, because taking plume edge points would lead to under- or overestimation, similar to the side view method. These measured lengths combined with the corresponding view zenith

angles lead to height estimates of ~8.2 km and ~7.9 km.

     For the special case of a vertical column, height can similarly be estimated from the true (stick) shadow length (method 2), which, however, is a more challenging technique in practice. Determining the exact location of the shadow terminus can be difficult due to the blurring effect of the penumbra, the lighter outer part of the shadow where the sun disk is only


partially blocked. For the GOES-17 near-limb view, this is compounded by the large GIFOV and severe map distortions. In this particular case, part of the true shadow is also obscured in the GOES-17 image by the eruption column itself, because the satellite is close to the principal plane and on the same side as the rising sun. Finally, the shadow of the eruption column is cast on a cloud layer rather than on the surface and hence corresponds to height above the cloud. This, however, is a relatively minor nuisance, because accounting for the height of a flat low-level cloud layer is no more error prone than the handling of the complex local topography would be under clear-sky conditions.

The height of the cloud layer, which we assume constant over the scene, can be estimated from the length (~10.3 km) and direction of its shadow. The cloud shadow is visible in the Himawari-8 image (marked by the double-headed white arrow) but invisible from the particular sun-view geometry of GOES-17. In order to avoid topography effects, we actually measured the cloud edge shadow over the ocean, northeast of the land shadow used for illustration in Fig. 4b. Applying method 3 to this edge shadow measurement yields a cloud top height (CTH) of ~1.2 km. The ERA5 profiles indicate a low-level

temperature inversion over the ocean, between 0.93 km and 1.36 km. Considering that marine stratus tends to be located near the base of the temperature inversion and has a typical thickness of 200–300 m, the CTH estimated by the edge shadow method seems reasonable.

The start point and end point of the *visible* plume shadow are marked by the green and yellow dots, respectively. The start point observed in the Himawari-8 image is the *actual* shadow start point, i.e. the vent location, which leads to an

observed (full) shadow length of ~77.7 km. In contrast, the beginning portion of the shadow (between the black triangle of the vent and the green dot in Fig. 4a) is obscured by the eruption column itself from the GOES-17 perspective and thus the observed shadow length is a shorter ~68.5 km. The distance between the vent and the shadow terminus (or deduced shadow length), however, is ~76.8 km in the GOES-17 image, similar to the one actually observed by Himawari-8. Plugging the Himawari-8 measured full shadow length and the corresponding solar zenith angle into Eq. (2) of Part 1, yields a plume

height of ~7.1 km above the cloud layer or, by adding the CTH, a height of ~8.3 km above the ellipsoid. Similarly, the full shadow length deduced from, but not entirely observed in, GOES-17 data translates to an above-ellipsoid height of ~8.0 km.

Here we note that the ellipsoid-projected location of the shadow terminus that would be observed under cloud-free conditions (and no surface topography) can be approximately determined from the sun-satellite geometry and the estimated CTH. This clear-sky shadow end (marked by a yellow triangle) can be obtained by (i) shifting the observed shadow end cast

on the cloud layer (yellow dot) in the direction opposite to the (map distorted) view azimuth by a distance of CTH times the tangent of the view zenith angle and then (ii) shifting this intermediate point in the solar azimuth direction by a distance of CTH times the tangent of the solar zenith angle. This backward and forward projection roughly removes the cloud height parallax, which is responsible for the significantly different image locations (yellow dots) of the cloud-top shadow end in GOES-17 and Himawari-8 data. The discrepancy between the estimated clear-sky shadow end image locations (yellow

triangles) is much smaller. Using the distance between the yellow triangles and the volcano base in Eq. (2) directly yields the height above the ellipsoid; in our case a distance of ~92.0 km results in a height estimate of ~8.2 km.



The edge shadow technique (method 3), which is the generalization of the previous stick shadow technique to a horizontally extensive suspended ash layer, can also be used as a separate estimate for a vertical eruption column. In this method, height is fixed by the length and direction of the line segment connecting the plume edge to the shadow terminus; 165 for a suspended ash layer this line segment is the apparent (observed) shadow, but for a narrow vertical column it actually runs through an unshadowed area (see section 3.3 in Part 1). As shown in Fig. 4, the distance between the westernmost edge point of the umbrella cloud (cyan dot) and the shadow terminus cast on the cloud layer (yellow dot) is ~37.6 km for GOES-17 and ~80.0 km for Himawari-8. For the sun-satellite geometry of the plotted images, these distances respectively correspond to an above-cloud height of ~6.4 km and ~6.6 km, or an above-ellipsoid height of ~7.6 km and ~7.8 km. As 170 before, considering the line segment between the plume edge and the estimated clear-sky shadow end (cyan dot to yellow triangle), directly yields similar above-ellipsoid height values, without the need to add the CTH.

Finally, method 3 can also be applied in 'stereo mode', using as input the GOES-17 and Himawari-8 image locations and sun-view geometry angles of the center of the umbrella cloud (blue dots in Fig. 4b). The ~80.0 km parallax between these two sensor-projected locations is unaffected by the presence of the low-level cloud layer and translates to an above-ellipsoid 175 height of ~8.0 km.

It is difficult to put exact error bars on these height estimates, but an uncertainty of a few hundred meters seems reasonable. The traditional geometric methods use the flat earth approximation, which is usually justified by evoking that the radius of earth is much greater than the length of the shadow or that of the map-projected image of the eruption column. Over the horizontal distances measured in Fig. 4 (max ~90 km), the GOES-17 view zenith angle and the solar zenith angle 180 can change by up to ~0.7º due to earth's curvature. Near a zenith angle of 84º, this change leads to a ~12% change in the tangent function used to convert distance to height. In our calculations, we used the average of the zenith angles at (i) the volcano base and (ii) the shadow terminus or sensor-projected plume top position, which is a 1$^{st}$ order correction for surface curvature. In fact, an advantage of the side view method, which operates in angular space, is that it avoids the non-linear sensitivity to the tangent function, which is a feature of the methods based on surface-projected distances. The discussed 185 difficulty in determining the shadow terminus, neglecting refraction effects on solar zenith angle, and spatial variations in the height of the low-level cloud layer represent additional uncertainties for the shadow-based estimates. Nevertheless, the height estimates from these alternative geometric techniques are consistent with the side view estimate and cluster around ~8±0.5 km.

The GOES-17 and Himawari-8 band 14, 11 μm brightness temperatures at the eruption time of 19:10 UTC are plotted in 190 Fig. 5a and Fig. 5b. The thermal infrared channel has a resolution of ~2 km at the subsatellite point. For the extensive low-level cloud layer around the volcano, the GOES-17 $BT_{11}$ (252.1–253.1 K) is 3–4 K lower than the Himawari-8 $BT_{11}$ (255.6–256.6 K) due to limb cooling caused by the increased optical pathlength of the absorbing atmosphere at oblique view angles. For the ash plume, however, the temperature contrast between the satellites is the opposite: the GOES-17 minimum $BT_{11}$ ('dark pixel', 231.2 K) is 5.4 K higher than the Himawari-8 minimum value (225.8 K). From the oblique side view 195 perspective of GOES-17, the plume fills a smaller fraction of the relatively large infrared pixels than from the less oblique



Himawari-8 view point. The larger beam-filling (or subpixel ash cloud) error over the plume leads to a more significant contamination of GOES-17 infrared data by the warmer lower-level cloud layer.

The plume heights corresponding to the dark pixel temperatures between 19:10–19:50 UTC are shown in Fig. 5c. For both satellites the minimum $BT_{11}$ decreases by 3–4 K over this time period; however, the GOES-17 temperatures remain 5–6 K warmer than the Himawari-8 ones. The decrease in $BT_{11}$ is due likely to the combination of a slight increase in plume height and a reduced beam-filling error as the plume expands, but the individual contributions of these two effects cannot be determined. Illustrating a common problem with the temperature method, the Himawari-8 minimum $BT_{11}$ leads to multiple solutions because of the temperature inversion at the tropopause in the ERA5 profile. At the eruption time the dark pixel temperature matches the ERA5 data at 7.6 km, 11.0 km, and 14.0 km, while later on there are two matching heights varying between 8.0–10.5 km. The VolSatView algorithm retrieves a height of ~9.6 km at 19:10 UTC, close to the average of the two lowermost solutions. The warm-biased GOES-17 dark pixel temperature results in a single but low-biased height between 6.8–7.2 km. Overall, the lowermost temperature-based heights derived from Himawari-8 data close to the eruption time are consistent with the 19:10 UTC geometric retrievals. A height of ~8 km indicates a plume that did not penetrate the cold point tropopause (~8.7 km), which is in accordance with the observed spreading of the umbrella cloud and the lack of overshooting tops.

### 3.2 Sheveluch, 10 April 2019

An explosive eruption of the volcano occurred on 10 April 2019 at 02:53 UTC with continuous ash removal until 04:25 UTC. KVERT issued an orange coded VONA (2019-77, http://www.kscnet.ru/ivs/kvert/van/?n=2019-77), reporting a plume height of 7.5–8.0 km at 03:13 UTC based on video surveillance. The GOES-17 and Himawari-8 visible imagery depicting the plume's evolution until 05:00 UTC are given in Supplementary Animation 2 (SA 2). The corresponding ground-based video footage at 1-minute cadence obtained from the webcam located at the Levinson-Lessing Kamchatkan Volcanological Station in Klyuchi (yellow square in SA 2; 56.3167ºN, 160.8333ºE), ~45 km southwest of Sheveluch, is given in Supplementary Animation 3. As shown, the eruption produced a weak bent plume without an umbrella cloud, which was quickly advected south by strong northerly winds; the ERA5 wind speed increased from 6.1 m s$^{-1}$ at the vent to a maximum of 19.5 m s$^{-1}$ at an altitude of 8.7 km. Plume height estimates obtained by the different geometric and temperature methods at the eruption time are summarized in Table 1.

The GOES-17 fixed grid image at 03:00 UTC is plotted in Fig. 6a with base-relative isoheights drawn. The height of the bent plume right above the vent is ~6 km. Because the plume expanded perpendicular to the line of sight, the maximum plume height can be determined with little error using plume top locations south of the vent. The plume top element marked by the blue dot yields an altitude of ~7.2 km. Identifying this plume top location in the equirectangular map (see SA 2) leads to a projected column length of ~66.0 km, which results in a height of ~7.1 km using method 1 with the GOES-17 VZA.

The Himawari-8 gridded and resampled image (CEReS V20190123) is shown in Fig. 6b, with the volcano base and the same plume top element also marked. In this case, a projected column length of ~16.3 km corresponds to a height of ~6.9 km



using method 1 with the Himawari-8 VZA. Finally, the ellipsoid-projected parallax between the GOES-17 and Himawari-8 plume top locations (blue dots) is ~70.0 km, which translates to a height of ~7.0 km using method 3 (shadow method in 'stereo' mode).

The Klyuchi webcam image of the plume at 03:00 UTC is shown in Fig. 6c. The volcano and lower parts of the plume were obscured by clouds from this vantage point. Webcam data are approximately calibrated by identifying in cloud-free images the location of known features, such as degassing and incandescence on the Young Sheveluch lava dome (~2.5 km elevation, black triangle) and the peak of Old Sheveluch (~3.3 km elevation), as well as by theodolite measurements of further image points. Note that the plume's movement or bending along the line of sight introduces height biases similar to those of the side view technique. The height reported in the VONA is based on expert analysis of video data from two cameras and tries to account for the particulars of the eruption, such as bending of the column. With this caveat in mind, expert analysis of the video data yielded an estimate of ~7 km for the plume top element marked in Fig. 6c.

The plume heights derived from the dark pixel temperatures between 03:00–04:00 UTC are shown in Fig. 6d. The red and blue vertical lines near 245–246 K respectively indicate the GOES-17 and Himawari-8 minimum $BT_{11}$ at 03:00 UTC. Due to the small size of the plume at that time, both brightness temperatures suffer from a large beam-filling error and are thus biased warm, leading to a height estimate of ~4.5 km that is significantly lower than the geometric height estimates. Ten minutes later, the GOES-17 and Himawari-8 minimum $BT_{11}$ exhibits a large decrease (by ~9 K and ~15 K), thanks to a considerable reduction in the beam-filling error over the rapidly expanding plume. Between 03:10–04:00 UTC the GOES-17 and Himawari-8 minimum $BT_{11}$ decreases from 236.4 K to 231.0 K and from 231.2 K to 226.6 K, respectively (red and blue shading in Fig. 6d). As before, the GOES-17 dark pixel temperatures are 4–5 K warmer than the Himawari-8 values, because of the comparatively larger beam-filling error in the oblique side view. These $BT_{11}$ values correspond to plume heights between 6.2–6.9 km for GOES-17 and 6.9–7.5 km for Himawari-8.

Using the U.S. Standard Atmosphere 1976 (ITU-R, 2017) instead of ERA5, which is an option in VolSatView, results in heights that are ~2 km higher, highlighting the sensitivity of the temperature method to the assumed atmospheric profile. At 03:00 UTC this leads to a better agreement with the geometric estimates, later on, however, the derived heights are considerable overestimates.

In sum, the temperature-based plume height is a severe underestimate at eruption time, but it catches up with the geometric heights, especially for Himawari-8, as the magnitude of the beam-filling error decreases with the spreading of the plume. In general agreement with our results, Bril et al. (2019) also found temperature-based heights underestimating stereo heights for small plume tops below the tropopause temperature inversion.

### 3.3 Sheveluch, 29-30 August 2019

On 29 August 2019 the volcano showed prolonged eruptive activity, which eventually resulted in a strong plume with a large umbrella cloud. The GOES-17 and Himawari-8 visible imagery between 01:00–07:00 UTC are given in Supplementary Animation 4 (SA 4). A high-level cloud layer at an altitude of ~8 km (7–9 km) obscured the satellite view of the volcano and



the lower part of the eruption column, while low-level clouds obscured the view of the entire plume from the Klyuchi webcam. Nevertheless, the satellite animation suggested explosions at least from 01:00 UTC. The rising eruption column pierced the high cloud layer at 01:50 UTC and by 02:50 UTC started to flatten and spread out to form an umbrella cloud of
rapidly increasing size. The ERA5 data indicated northwesterly winds at lower levels and southeasterly winds at higher levels, with wind speed increasing from 5 m s$^{-1}$ at the vent to a maximum of 11.5 m s$^{-1}$ at an altitude of 9.5 km.

KVERT issued an orange coded VONA (2019-125, http://www.kscnet.ru/ivs/kvert/van/?n=2019-125), reporting a plume height of 9.0–10.0 km at 03:10 UTC based on Himawari-8 brightness temperatures. The corresponding GOES-17 fixed grid image (with base-relative isoheights) and the Himawari-8 CEReS V20190123 image are shown in Figs. 7a and 7b. The
umbrella cloud can be fitted well with a circle of ~20 km radius. Using the center of the circle as a representative plume top location yields a height of ~10.8 km by the side view method. An overshooting top at an altitude of 12.0–12.5 km is also apparent at this time. Later images show overshooting tops and, thus, individual explosions, of diminishing magnitude occurring until 04:30 UTC (see SA 4). The umbrella cloud keeps spreading at ~11 km altitude with the clearly visible propagation of circular shock waves and becomes increasingly elongated in the downwind northwesterly direction. By 04:30
UTC the still circular upwind side of the anvil cloud expands to a radius of ~50 km.

The variation of the minimum BT$_{11}$ between 02:00–04:00 UTC and the ERA5 temperature profile at 03:00 UTC (which is representative of this time period) are plotted in Fig. 7c. The beam-filling error (subpixel effect) is negligible over the large umbrella cloud and hence the GOES-17 brightness temperatures (219.0–223.2 K) are 1–2 K lower than the Himawari-8 values (220.0–223.8 K) due to limb cooling. The reanalysis profile shows a sharp inversion at the tropopause cold point of
221.9 K near an altitude of 10.3 km. As a result, the warmer end of the measured BT$_{11}$ range leads to multiple height solutions: lower values between 9–11 km and higher values in a wide range between 16–26 km. After 02:40 UTC, however, the measured dark pixel temperatures indicate undercooling below the ERA5 cold point and thus yield no height solution at all.

These dark pixel temperatures can technically be matched to an altitude if a standard model atmosphere with a colder
tropopause is used instead, although such a model profile is a less accurate representation of the actual temperature structure than ERA5. For example, considering the U.S. Standard Atmosphere 1976 used in VolSatView, the GOES-17 and Himawari-8 dark pixel temperature at 03:10 UTC correspond to a plume height of ~10.6 km and ~10.4 km, respectively, in apparently good agreement with the side view estimate of ~10.8 km. Higher estimates (~12.4 km) would be obtained using the mid-latitude summer atmosphere, but the putatively more appropriate sub-arctic summer atmosphere with a tropopause at
225 K would result in plume undercooling and no height solution, similar to the ERA5 profile. Overall, this case demonstrates again the sensitivity of the temperature method to the assumed atmospheric profile.

Another weaker explosion occurred on 30 August 2019 at 08:00 UTC, for which KVERT also issued an orange coded VONA (2019-128, http://www.kscnet.ru/ivs/kvert/van/?n=2019-128). This case is just before sunset and thus provides a good demonstration of the side view method in a low-light environment. The eruption produced a weak plume without an
umbrella cloud, which was bent and then advected south-southeast by strong north-northwesterly winds; the ERA5 wind





speed increased from 6 m s$^{-1}$ at the vent to a maximum of 20 m s$^{-1}$ at an altitude of 10.5 km. The plume is generally similar to the 10 April 2019 case (see section 3.2 and Fig. 6). As before, height estimates obtained by the various methods are summarized in Table 1.

The contrast-enhanced GOES-17 fixed grid image is plotted in Fig. 8a. The plume is close to the day-night terminator and GOES-17 views its shadowed eastern side. Nevertheless, the shape of the ash column, bent perpendicular to the line of sight, can be made out fairly well in the image. The plume top element marked by the blue dot indicates an altitude of ~7.2 km. Identifying this plume top feature in an equirectangular map (not shown) leads to a projected column length of ~67.8 km, which results in a height of ~7.3 km using method 1 with the GOES-17 VZA.

The contrast-enhanced Himawari-8 gridded image is shown in Fig. 8b, with the volcano base and the same plume top
element also marked. The lighting conditions are more favorable here, because the satellite observes the plume's western side illuminated by the setting sun. In this case, a projected column length of ~16.7 km corresponds to a height of ~7.0 km using method 1 with the Himawari-8 VZA. Finally, the ellipsoid-projected parallax between the GOES-17 and Himawari-8 plume top locations (blue dots) is ~68.7 km, which translates to a height of ~6.8 km using method 3 (shadow method in 'stereo' mode).

The corresponding Klyuchi webcam image of the plume is shown with altitude markings in Fig. 8c. Analysis of the video data, considering the bending of the column and the partial obscuration of the plume top by meteorological clouds, resulted in a height estimate of 7.0–7.5 km. In sum, the satellite and ground-based geometric estimates are fairly consistent and indicate a plume height near 7 km.

The dark pixel brightness temperatures between 08:00–09:00 UTC and the ERA5 temperature profile at 08:00 UTC are
plotted in Fig. 8c. The beam-filling error is generally substantial for this relatively small plume, but it has a larger effect on Himawari-8 $BT_{11}$ than GOES-17 $BT_{11}$, because the plume appears narrower from the former's perspective due to the combination of plume morphology and view geometry. As a result, the GOES-17 minimum $BT_{11}$ (253.4–257.6 K) is 1–2 K colder than the Himawari-8 values (254.8–259.5 K). These temperature ranges correspond to respective height ranges of 5.0–5.6 km and 4.7–5.4 km with the ERA5 profile. For reference, using the U. S. Standard Atmosphere would lead to 0.3 km
lower heights. These results again demonstrate that the temperature-based height is a considerable underestimate for small and thin plumes. For an eruption with sustained and larger ash release, such as the 10 April 2019 event, the temperature method can catch up with the geometric estimates as the plume spreads and optically thickens. This, however, is not the case for the current short duration puff, whose plume quickly dispersed.

### 3.4 Karymsky, 14 August 2019

The volcano erupted on 14 August 2019, shortly before 04:30 UTC. The GOES-17 and Himawari-8 visible imagery between 04:00–06:00 UTC are plotted in Supplementary Animation 5 (SA 5). The vicinity of the volcano is covered by a mid-level cloud layer at an altitude of 4.0–4.5 km, while further inland Cu and Cb clouds are developing. In fact, at 05:40 UTC the GOES-17 fixed grid image (top left corner) shows a textbook side view of a faraway thundercloud with an overshooting top





and anvil. At 04:30 UTC, the eruption plume breaks through the mid-level clouds and quickly dissipates without forming an
umbrella. The ERA5 data indicate a wind speed of 5 m s$^{-1}$ at the vent and 8 m s$^{-1}$ at an altitude of 10 km, with north-northeasterly winds at lower levels and west-northwesterly winds at higher levels. The plume splits in two with the upper main part advecting east and the smaller lower part advecting south.

KVERT issued an orange coded VONA (2019-116, http://www.kscnet.ru/ivs/kvert/van/?n=2019-116), reporting a plume height of 4.0–4.5 km at 04:45 UTC based on visual data by nearby volcanologists. The GOES-17 fixed grid image and the
Himawari-8 image at 04:30 UTC are shown in Figs. 9a and 9b. Two plume top peaks can be clearly identified in the satellite images, marked by the light and dark blue dots. The side view technique indicates both peaks at an altitude of ~7.1 km. For the peak closer to the vent (light blue dot, minimal bending), the projected length is ~66.0 km and ~14.5 km in the GOES-17 and Himawari-8 mapped image, respectively, which yields a height of ~6.8 km and ~7.0 km using method 1. The ellipsoid-projected parallaxes between the GOES-17 and Himawari-8 plume top locations are ~68.7 km (light blue dots) and ~73.4 km
(dark blue dots), which translate to height values of ~6.9 km and ~7.3 km using 'stereo' method 3.

The photo of the lower part of the plume obtained by a quadcopter at 04:45 UTC is shown in Fig. 9c. Obviously, the quadcopter image only captures the plume below the cloud layer and hence provides a height underestimate. Considering that the summit of the volcano is at an elevation of ~1.5 km, a rough visual estimate from the photo results in a plume height of 4.0–4.5 km, as reported in the VONA.

The variation of the minimum BT$_{11}$ between 04:00–06:00 UTC and the ERA5 temperature profile at 05:00 UTC are plotted in Fig. 9d. Because the plume is small, the brightness temperature is likely affected by substantial beam-filling error for both satellites. However, while Himawari-8 observes the plume against the warmer background of the mid-level cloud layer surrounding the volcano, GOES-17 observes the plume partially against the background of the convective clouds developing further inland in the west. Combined with limb cooling, this leads to the GOES-17 dark pixel BT$_{11}$ (253.3–256.5
K) being 2–3 K colder than the Himawari-8 value (256.8–259.1 K). These temperature ranges correspond to respective height ranges of 5.7–6.2 km and 5.3–5.7 km using the ERA5 profile; the heights are ~0.8 km lower when using the U. S. Standard Atmosphere in VolSatView. In sum, the geometric methods consistently indicate a plume height of ~7 km, while the temperature-based heights are biased low by at least 1 km, in line with our previous findings for small and thin plumes.

### 3.5 Bezymianny, 21 October 2020

A detailed description of this case is provided by Girina et al. (2020), here we focus only on the salient early features of the eruption. KVERT issued red coded VONAs (2020-190, http://www.kscnet.ru/ivs/kvert/van/?n=2020-190 and 2020-191, http://www.kscnet.ru/ivs/kvert/van/?n=2020-191), reporting a plume height of 8.0–9.0 km based on video data and Himawari-8 brightness temperatures. According to video surveillance from the Kirishev seismic station, located 16 km west-southwest of Bezymianny and operated by the Kamchatka Branch of the Geophysical Survey of the Russian Academy of
Sciences, the explosive eruption of the volcano started at 20:22 UTC, just around sunrise. The ground-based video of the first 30 minutes of the eruption is available at https://www.youtube.com/watch?v=LFOSJtGRON8, while the GOES-17 and





Himawari-8 visible images between 20:00–23:00 UTC are given in Supplementary Animation 6. Note that the GOES-17 view (azimuth of 112º) and the Himawari-8 view (azimuth of -156º) are almost exactly perpendicular to each other, resulting in a rather different appearance of the early minimally-expanded eruption column. At 20:50 UTC the overshooting top of the

ash column, which penetrated the stratosphere, is very prominent in the GOES-17 side view but difficult to discern in the Himawari-8 image. The plume eventually split into three parts: the lowest part expanded towards north-northwest, a higher lobe advected south-southeast, and the highest part that directly originates from the overshooting top drifted east due to relatively modest winds of 5–11 m s$^{-1}$.

The GOES-17 fixed grid image and the Himawari-8 image at 20:30 UTC are shown in Figs. 10a and 10b, while the

Kirishev webcam image at 20:29 UTC is plotted in Fig. 10c (by 20:30 UTC the plume top is out of frame). The plume top location is indicated by the yellow and blue asterisks in the satellite images. The side view technique yields a peak height of ~9.3 km. The projected length of the eruption column is ~89.6 km and ~19.8 km in the GOES-17 and Himawari-8 mapped image, respectively, which converts to a height of ~9.4 km and ~8.7 km using method 1. The ellipsoid-projected parallax between the GOES-17 and Himawari-8 plume top locations is ~90.1 km, which translates to a height of ~9.1 km using

'stereo' method 3. The plume height estimated from the webcam image, considering the elevation of the nearby Klyuchevskoy (~4.8 km) and Kamen (~4.6 km) volcanoes, is ~9 km.

By 20:40 UTC (Figs. 10d and 10e), the column rises further and starts to develop an umbrella, spreading out mostly in the SW–NE direction. The side view method indicates a peak height of ~13.2 km. The projected lengths (~133.5 km and ~28.4 km) yield height estimates of ~13.5 km and ~12.4 km for GOES-17 and Himawari-8, respectively, while the parallax

(~134 km) results in a top height of ~13.1 km.

At 20:50 UTC (Figs. 10f and 10g), the overshooting top of the eruption column reaches its maximum altitude of ~15.3 km according to the side view technique. The height estimates based on the projected lengths (~158.5 km and ~35.7 km) are 15.7 km and 15. 5 km for GOES-17 and Himawari-8, respectively, and the parallax (~159.5 km) corresponds to a height of ~15.2 km.

The variation of the dark pixel $BT_{11}$ between 20:30–22:30 UTC and the ERA5 temperature profile at 21:00 UTC are plotted in Fig. 10h. The GOES-17 brightness temperatures are 2–3 K warmer than the Himawari-8 ones, due likely to the differing interplay between plume morphology, view geometry, and the background against which the plume is observed. The red (238.5 K) and blue (235.3 K) vertical lines respectively indicate the GOES-17 and Himawari-8 minimum $BT_{11}$ at 20:30 UTC. Because of the initially small size of the plume, these temperature values are biased warm, leading to low-biased

height estimates of 6.4 km and 6.8 km with the ERA5 profile, or 7.7 km and 8.1 km with the U.S. Standard Atmosphere used by VolSatView. Ten minutes later, as the plume rapidly expands and the beam-filling error decreases, the $BT_{11}$ drops by ~15 K and remains at values that correspond to dual height solutions fluctuating within a wide range. The ERA5 temperature profile is characterized by a cold point tropopause (216.6 K) at 10.1 km and a thick nearly isothermal (220–221 K) layer between 11.0–20.0 km. As a result, the dark pixel $BT_{11}$ after 20:30 UTC yields below-tropopause heights between 8.5–9.3



km and above-tropopause heights between 11.0–20.0 km. VolSatView avoids the issue of multiple solutions, at the expense of using a less accurate standard temperature profile, and retrieves a plume height near ~10.7 km.

     Overall, this case highlights again the good agreement between the geometric methods, but also the potential problems affecting the temperature method: low-biased heights for small and thin plumes as well as retrieval difficulties associated with the tropopause temperature inversion and small lapse rates (multiple solutions, excessive uncertainty).

**3.6 Ulawun, 26 June 2019**

     The volcano experienced a major subplinian eruption on 26 June 2019, which triggered an aviation warning with a red color code. Supplementary Animation 7 shows the GOES-17 and Himawari-8 visible images depicting the explosive activity between 04:00–07:50 UTC. Ulawun is observed by GOES-17 from almost exactly east (view azimuth of ~88º) at a view zenith angle of ~80º (Fig. 1). Himawari-8, on the other hand, observes the volcano from the northwest (view azimuth of

about -64º) at a small view zenith angle of ~14º. Thanks to the nearly overhead view, the height parallax is small in the Himawari-8 images. Also note that the equatorial location (5.05ºS, 151.33ºE) results in negligible distortion in the mapped view and solar azimuth directions.

     Explosions of varying strength and the peculiarities of the wind profile produced a multilayer plume, the highest part of which penetrated the stratosphere. The first major pulse at 04:40 UTC attained an altitude of 14–15 km according to the side

view method, with its plume spreading south-southeast by north-northwesterly winds, which were present between 10–15 km and showed a local maximum (~13.5 m s$^{-1}$) at 14 km altitude in the ERA5 profile. Note that this eruption column can be clearly seen in the GOES-17 oblique view already at 04:40 UTC when its top has a radius of merely 5 km, but the plume is difficult to distinguish from the surrounding meteorological clouds in the Himawari-8 overhead view, where the plume only becomes apparent in subsequent images as its umbrella grows.

Another series of pulses started at 05:20 UTC, which produced overshooting tops between 16–18 km. The umbrella formed by these puffs is darker than that of the 04:40 UTC puff, but it also expanded mostly to the south and at the same 14–15 km level. The strongest pulse began at 06:00 UTC, whose overshooting top reached its maximum altitude at 06:10 UTC; the corresponding GOES-17 and Himawari-8 images are plotted in Figs. 11a and 11b, respectively.

     The umbrella cloud of the pulses starting at 05:20 UTC can be fitted with a circle of ~28 km radius at 06:10 UTC. The

center of this fitted circle (yellow plus sign), located slightly southeast of the volcano, is at a height of 14.7 km, while the highest point of the overshooting top (yellow asterisk) is at an altitude of ~22.3 km, according to the side view technique. The projected length of the overshooting peak is ~147.1 km in the GOES-17 mapped image, which yields a height of ~21.5 km using method 1. The parallax between the GOES-17 and Himawari-8 overshooting top locations is ~149.7 km, which translates to a height of ~21.3 km using method 3 in 'stereo' mode.

The bulk of the 06:10 UTC puff spreads out in a highly isotropic manner, forming a rapidly growing and almost perfectly circular umbrella cloud above the previous southwardly elongated ash layer. Note the outwardly propagating concentric gravity waves in the animation. As plotted in Figs. 11c and 11d, the radius of the upper-level umbrella increases to ~63 km



by 07:00 UTC (the radius eventually reaches ~95 km by sunset at 07:50 UTC). The center of the fitted circle, which is horizontally very close to the volcano indicating little advection, corresponds to an altitude of ~18.3 km by the side view method. The shadow this upper-level umbrella casts on the lower-level ash cloud is ~18.8 km long in the azimuth direction of ~115º (Fig. 11d), which results in a ~3.2 km height differential between the layers using method 3. This shadow-based height difference estimate is consistent, within error bars, with the side-view plume height estimates (~14.7 km and ~18.3 km). The isotropic expansion of the upper umbrella cloud at ~18 km altitude is aided by a local wind speed minimum (~6.5 m s$^{-1}$) at 17 km altitude in the ERA5 data. Finally, there is an even higher plume element that directly originates from the overshooting top. The eastward movement of this smaller ash mass and its estimated height (~22.3 km) are again consistent with the ERA5 wind profile, which shows strong westerlies in the 19–24 km altitude range with a maximum speed of ~19.5 m s$^{-1}$.

The variation of the minimum $BT_{11}$ between 05:50–07:50 UTC and the representative ERA5 temperature profile at 07:00 UTC are plotted in Fig. 12. The tropical standard atmosphere profile (ITU-R, 2017) is also shown, which is close to the reanalysis data up to an altitude of 19 km and thus leads to similar derived heights. The GOES-17 dark pixel temperatures (191.9–197.0 K) are 4–5 K warmer than the Himawari-8 values (187.7–193.8 K), despite limb cooling. A potential reason for this temperature contrast is that the longest optical path in the nearly overhead Himawari-8 view samples almost the entire vertical extent of the tall eruption column, while the oblique GOES-17 side view only samples the shorter horizontal cross-section of the narrow column. Furthermore, the thickest plume core is observed against the background of the high and cold ash umbrella in the near-nadir Himawari-8 image, but it is observed partially against the background of lower and warmer meteorological clouds in the GOES-17 side view.

The Himawari-8 $BT_{11}$ typically has no corresponding height solution, as it indicates undercooling below the 192.8 K cold point of the ERA5 tropopause located at 16.7 km. The warmest Himawari-8 temperatures and most of the GOES-17 temperatures, on the other hand, result in two height solutions, one below and one above the tropopause, ranging between 15.8–18.1 km. The higher solution agrees well with the geometric height estimate of the upper umbrella cloud. The prominent overshooting top, however, is not captured by the basic temperature retrievals.

Here we note that the more sophisticated multichannel VOLcanic Cloud Analysis Toolkit (VOLCAT; Pavolonis et al., 2013), retrieves height values for the Ulawun plume in the 16–20 km range with a maximum height of 22 km (Bachmeier, 2019). The heights for the upper umbrella cloud are generally consistent with our estimate. The maximum VOLCAT height is obtained around 07:00 UTC for the eastward-moving ash mass originating from the 06:10 UTC overshooting top and agrees well with the side view height estimate (~22.3 km) calculated for the overshooting top itself. At 06:10 UTC the overshooting top directly above the volcano is sub-pixel in the IR channels and its height is significantly underestimated by VOLCAT. By 07:00 UTC, however, part of the overshooting mass drifts eastward and expands, thereby reducing beam-filling biases in the brightness temperatures and the derived heights.



## 460  3.7 Raikoke, 21-22 June 2019

A series of nine major explosions that produced a very large and dense ash plume occurred on 21-22 June 2019, with minor emissions continuing for a few more days. KVERT issued three red coded VONAs (2019-100, http://www.kscnet.ru/ivs/kvert/van/?n=2019-100; 2019-101, http://www.kscnet.ru/ivs/kvert/van/?n=2019-101; 2019-102, http://www.kscnet.ru/ivs/kvert/van/?n=2019-102), reporting a plume height of 10.0–13.0 km based on Himawari-8
brightness temperatures. Some 40 aircraft were consequently diverted away from the plume (Crafford and Venzke, 2019). The GOES-17 and Himawari-8 visible images capturing the explosions between 17:50 UTC on 21 June and 08:50 UTC on 22 June are given in Supplementary Animation 8. The low levels were characterized by an anti-cyclonic circulation centered on Ekarma island a bit northeast, leading to weak southeasterly winds in the boundary layer near Raikoke. The high-level flow, however, was westerly and quite strong, with maximum wind speeds of 25–30 m s$^{-1}$ at 11 km altitude, which resulted
in the rapid eastward drift of the ash cloud.

Plume height estimates for all major explosive events are summarized in Supplementary Figs. S1 and S2 and Supplementary Table S1, here in Fig. 13 we only show four typical snapshots (two for each day). At 19:40 UTC on 21 June, the narrow eruption column is minimally bent and has a sharp tip (Figs. 13a and 13b). The plume height estimate from the GOES-17 side view is ~9.9 km. The respective GOES-17 and Himawari-8 map-projected lengths (method 1) yield heights of
~9.7 km and ~10.0 km, using the corresponding view zenith angles (~86° and ~57°). The parallax between the plume top positions in the satellite images (yellow asterisks) results in a height of ~10 km (method 3 in 'stereo' mode). The Himawari-8 view also captures the plume's shadow, the end of which is cast on the surrounding marine Sc deck. This allows employing the stick shadow and edge shadow methods as well, which indicate above-cloud heights of ~8.9 km and ~9.1 km. Considering that ERA5 shows a low-level temperature inversion between 365–795 m and that parts of neighboring Rasshua
(peak elevation 956 m) and Shiashkotan (peak elevation 944 m) islands are still visible, a cloud top height of ~700 m seems reasonable for the marine Sc. Added to the above-cloud heights, this leads to above-ellipsoid heights of ~9.6 km and ~9.8 km from the shadow methods. The GOES-17 and Himawari-8 dark pixel BT$_{11}$ (254.0 K and 251.1 K), on the other hand, have a warm bias over the narrow plume and correspond to significantly lower heights of only 6.1 km and 6.5 km (Fig. 13i). The VolSatView height of 8.8 km is in better agreement with the geometric estimates, but it is still an underestimate.
The eruption column at 22:00 UTC on 21 June is still near-vertical but more massive, sporting a flat and circular umbrella cloud of ~7 km radius (Figs. 13c and 13d). The umbrella is not yet distorted or advected by the wind and thus its center is at the island. The side view method retrieves a height of ~11.3 km for the centerline of the umbrella cloud, which is in good agreement with the ~11.6 km stereo height obtained from the GOES-17 and Himawari-8 image locations of the plume center (yellow asterisks). These estimates are near the 11.2 km altitude of the ERA5 cold point tropopause (Fig. 13i).
The Himawari-8 dark pixel BT$_{11}$ (222.5 K) is close to the temperature of the quasi-isothermal atmospheric layer spanning between 10–25 km. As a result, the temperature method ambiguously yields four matching heights at 10.7 km, 12.2 km, 13.7 km, and 23.7 km; although the average of the two lowermost solutions, perhaps fortuitously, agrees well with the geometric





height estimates. The warm-biased GOES-17 dark pixel temperature (228.3 K), however, corresponds to a low-biased height of ~9.4 km. The VolSatView height of 10.0 km is also biased low by more than a kilometer.

At 03:50 UTC on 22 June, the plume consists of an overshooting column and a larger umbrella cloud of ~13 km radius (Figs. 13e and 13f). Because the overshooting top is located approximately above the volcano, the side view method can be used with more confidence here. The side view estimate of ~15.6 km is consistent with the stereo height of ~15.8 km obtained for the midpoint of the overshooting top (yellow asterisks). The umbrella cloud, however, has clearly been advected east (towards GOES-17) by the strong upper level winds. Therefore, taking the plume center as the characteristic point in the

side view calculations would underestimate the height. The windward western edge of the umbrella is a better choice as it is closer to the local vertical at the volcano. Allowing for the ambiguity in delineating the western plume edge in the GOES-17 image, we estimate an umbrella height between 13.0–14.0 km. This height range is consistent with the stereo height of ~13.7 km derived for a specific umbrella cloud feature that is visually identifiable in both satellite images (blue asterisks). The Himawari-8 and GOES-17 minimum brightness temperatures are very similar for this thick expanded plume (222.6 K and

222.3 K) and again lead to four possible height solutions (Fig. 13j); the VolSatView height (10.1 km) is near the lowest of these. None of them is a good match for the height of the overshooting top, but the second highest solution around ~14 km agrees well with the geometric height estimates of the umbrella cloud.

The last major eruptive puff at 05:30 UTC on 22 June produced a wide and near-vertical overshooting column (Figs. 13g and 13h). The side view height estimate of ~14.5 km is in good agreement with the stereo height of ~14.6 km calculated for

the midpoint of the plume top (yellow asterisks). As before, the Himawari-8 dark pixel temperature (222.5 K) corresponds to four matching heights, the closest of which (13.4 km) underestimates the geometric heights by more than 1 km (Fig. 13j). The VolSatView result and the estimate based on the warm-biased GOES-17 minimum $BT_{11}$ (227.0 K) lead to even more underestimated heights of 10.2 km and 9.7 km, respectively.

In a final example, we analyze the plume at the overpass time of the *Terra* satellite carrying the Moderate Resolution

Imaging Spectroradiometer (MODIS). As described in section 3.4 of Part 1, the novel "3D Winds" algorithm performs stereo wind and height retrievals by combining near-simultaneous imagery from MODIS on one hand with that from GOES-17 or Himawari-8 on the other (Carr et al., 2019). The GOES-17 near-limb imagery is unsuitable for *automated* image matching— although it can occasionally be used for stereo matching by a human observer as demonstrated in our study—, but Himawari-8 imagery, characterized by less extreme view zenith angles over Raikoke, can be successfully combined with

MODIS imagery.

The GOES-17 and Himawari-8 views of the plume in the vicinity of the volcano at 01:20 UTC on 22 June are shown in Figs. 14a and 14b. The overall plume stretches hundreds of kilometers to the east at that time, because it has been formed by a train of quasi-continuous explosions starting at 22:40 UTC on 21 June and ending at 01:50 UTC on 22 June. The plume has a complex topography with large height variations and the side view method can only determine the height of the

overshooting top above the vent, which appears to reach an altitude of ~16.5 km. In good agreement with this estimate, the



simplified shadow-stereo technique (method 3 in stereo mode) yields a height of ~16.1 km for the northern edge of the overshooting top (yellow asterisks).

The "3D Winds" stereo plume heights derived from Himawari-8 band 3 and MODIS-*Terra* band 1 (0.64 μm, 250 m nominal resolution) images are plotted in Fig. 14c. The Himawari-8 fixed grid image was first remapped to the MODIS

granule after navigation adjustments per Yamamoto et al. (2020) and then retrievals were performed using 48×48 MODIS pixel ($12\times12$ km$^2$) image templates. As indicated by the white areas (missing data), stereo matching sometimes failed due to lack of texture or the different view azimuths (northwesterly for MODIS and southwesterly for Himawari-8). The former mostly affects the flatter and fuzzier downstream parts of the plume, while the latter can hamper retrievals over the more dynamic near-vent areas where the satellites might view different sides of the vertically protruding overshooting column.

The automated stereo algorithm shows a maximum plume height of ~15.5 km in the vicinity of the volcano, which is reasonable given that the highest part of the overshooting top is not captured due to reasons discussed above. The retrievals also reveal a substantial spatial gradient whereby the plume height decreases from 14–15 km in the northwest to 7–8 km in the southeast. Such height variation is qualitatively consistent with the clearly layered appearance of the plume in the satellite images. Supplementary Fig. S3 plots the "3D Winds" heights derived for the MODIS-*Aqua* overpass at 03:10 UTC

on 22 June, by which time the plume expanded meridionally and drifted further to the east. The mean plume height also increased as evidenced by a larger area fraction above 12 km, but the overall height range (7–15 km) and the south-north height gradient remained the same.

The corresponding ERA5 temperature profile and minimum brightness temperatures are plotted in Fig. 14d. The Himawari-8 dark pixel temperature (216.2 K) indicates a ~5 K undercooling even relative to the standard atmosphere and

yields no height estimate, while VolSatView returns an estimate of 11.1 km for the umbrella. The warmer GOES-17 value (221.3 K), on the other hand, corresponds to three ambiguous solutions (11.3 km, 14.7 km, and 22.5 km). These results demonstrate again the typical limitations of the temperature method.

## 4. Summary and outlook

The side view height estimation method introduced in Part 1 was tested on several recent volcanic plumes that were

observed near the limb at very oblique angles by GOES-17 but were viewed under favorable conditions by Himawari-8. Height estimates from the new method were compared to independent geometric estimates derived from shadows, stereo observations, and ground-based video footage as well as to brightness temperature-based height estimates. Although the number of test cases is relatively small—a total of seven eruptions were analyzed—and more validation is certainly needed, a few conclusions have already emerged from the results.

The side view heights were typically within 500 m of the other geometric height estimates, including expert-derived plume heights from nearby video camera images, building confidence in the technique. Somewhat surprisingly, height estimates could also be calculated from GOES-17–Himawari-8 stereo observations, which also agreed well with the side



view estimates. Near-limb imagery is generally unsuitable for automated stereo retrievals, because the standard area-based image matchers are unable to correlate such extreme views with less oblique ones. A human observer, however, can build a

3D mental model of the plume and identify common features (tops, edges, striations) even in images taken from vastly different perspectives. The "3D Winds" automated stereo algorithm, one of the comparison methods in this study, has recently been updated to handle geostationary satellite pairs too (Carr et al., 2020), but its coverage is limited to overlap areas within 65° of the subsatellite points and thus excludes our case study volcanoes. Extending the code's functionality to allow the manual specification of stereo point pairs between GOES-17 near-limb views and Himawari-8 images might be

worth the effort, given the positive experience gained in our study.

The temperature method, although irreplaceable in routine operations, demonstrated its well-known biases. For weak plumes or for the initial phases of strong plumes, when the eruption column is small and narrow, temperature-based heights can underestimate geometric heights by 2–3 km. Such subpixel plumes only partially fill the relatively large (2 km at the subsatellite point) thermal infrared field of view, which usually leads to warm-biased brightness temperatures and low-biased

heights due to radiance contribution from the surface or lower-level clouds. Such subpixel biases are reduced in the later phases of strong eruptions that develop large and optically thick umbrella clouds. Unlike the side view method, however, the temperature method hardly ever captured the overshooting tops, as these features usually remained (infrared) subpixel-scale even in strong plumes.

The temperature method also often yielded ambiguous results. When the plume temperature falls within the narrow

temperature range of the quasi-isothermal layer above the tropopause, multiple height solutions within a wide (10+ km) altitude range are possible. Although one of the matching heights sometimes agreed well with the geometric umbrella height, it was impossible to select the right solution a priori. Besides, temperature-based heights are generally sensitive to the assumed atmospheric temperature profile. Three of the investigated eruptions produced undercooled plumes at certain times, even precluding the application of the temperature method. The results also show that despite limb cooling, the GOES-17

dark pixel temperature can be warmer than the Himawari-8 value. The minimum temperature measured over the plume is an intricate function of view direction, plume size and 3D morphology, and the temperature of the background against which the plume is observed, further complicating the interpretation of temperature-based height retrievals.

We believe the presented case studies have demonstrated that GOES-17 near-limb observations, either independently or in stereo with Himawari-8 views, can be a useful complementary to the more established height retrieval methods. The study

of Bruckert et al. (2021) in the current issue, for example, has found that the better characterization of the height and timing of the explosive puffs during the 2019 Raikoke eruption, enabled by GOES-17 oblique imagery, improves the simulation of the temporal evolution of total atmospheric ash content. Constraining the source terms in the modeling of near-field plumes might indeed be the most promising application of the side view height retrieval technique.



*Data availability*. The GOES-R ABI L1B radiances are available from the NOAA Comprehensive Large Array-data
Stewardship System (CLASS) archive (https://www.avl.class.noaa.gov/saa/products/welcome). Himawari 8/9 gridded data
are distributed by the Center for Environmental Remote Sensing (CEReS), Chiba University, Japan (http://www.cr.chiba-
u.jp/databases/GEO/H8_9/FD/index.html). The Holocene Volcano List is compiled by the Global Volcanism Program:
Volcanoes of the World, v. 4.9.1. Venzke, E (ed.), Smithsonian Institution, https://doi.org/10.5479/si.GVP.VOTW4-2013.
The KVERT Volcano list is available at http://www.kscnet.ru/ivs/kvert/volcano.php?lang=en. VolSatView uses the
following dataset: National Centers for Environmental Prediction/National Weather Service/NOAA/U.S. Department of
Commerce, European Centre for Medium-Range Weather Forecasts, and Unidata/University Corporation for Atmospheric
Research: Historical Unidata Internet Data Distribution (IDD) Gridded Model Data, Research Data Archive at the National
Center for Atmospheric Research, Computational and Information Systems Laboratory, https://doi.org/10.5065/549X-KE89,
2003. The PeakVisor summit database (freely searchable on the website) and mobile app (requiring a subscription) are
available at https://peakvisor.com.

*Author contributions*. ÁH developed the idea and methodology of the side view retrievals during discussions with GAH and
SAB. Retrievals from the 3D Winds stereo code were provided by its developers JLC and DLW, while retrievals from the
VolSatView information system were provided by its creators AAB, AAM, OAG, and DVM. ÁH analyzed the results and
prepared the manuscript with significant contributions from all authors.

*Competing interests*. The authors declare that they have no conflict of interest.

*Special issue statement*. This article is part of the special issue "Satellite observations, in situ measurements and model
simulations of the 2019 Raikoke eruption (ACP/AMT/GMD inter-journal SI)". It is not associated with a conference.

*Acknowledgements*. ÁH, GAH, and SAB are members of the VolPlume project within the research unit VolImpact funded by
the German Research Foundation DFG (FOR 2820). This work also contributes to the Cluster of Excellence "CLICCS—
Climate, Climatic Change, and Society" funded by the Deutsche Forschungsgemeinschaft DFG (EXC 2037, Project Number
390683824), and to the Center for Earth System Research and Sustainability (CEN) of Universität Hamburg.

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





**Figure 1.** GOES-17 (solid red) and Himawari-8 (dashed blue) view geometry for our case study volcanoes (triangles). View zenith angle for **(a)** Sheveluch (S), Bezymianny (B), Karymsky (K), and Raikoke (R), and **(c)** Ulawun (U). The corresponding view azimuth angle measured clockwise from north (+) for GOES-17 and anticlockwise from north (-) for Himawari-8 is given in panels **(b)** and **(d)**.







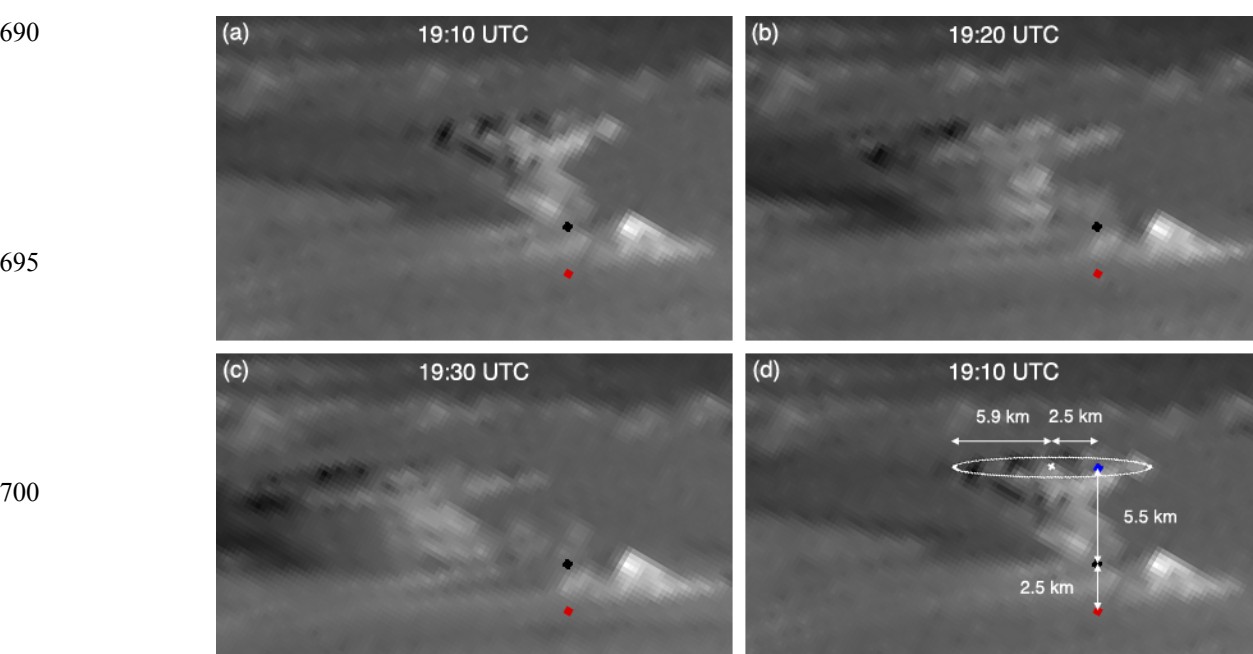


**Figure 2.** GOES-17 band 2 fixed grid image (8x magnification) of the 8 April 2020 eruption of Sheveluch at **(a)** 19:10 UTC, **(b)** 19:20 UTC, **(c)** 19:30 UTC, and **(d)** 19:10 UTC with plume geometry indicated. The images were rotated clockwise by the geodetic colatitude angle; south is left, north is right, west is up, east is down. The red and black diamonds respectively mark the base and vent of Young Sheveluch. In panel **(d)**, the umbrella cloud is fitted with a circle of 5.9 km radius (distorted into an ellipse in the fixed grid view) and the blue diamond is our best visual estimate plume top position above the vent.






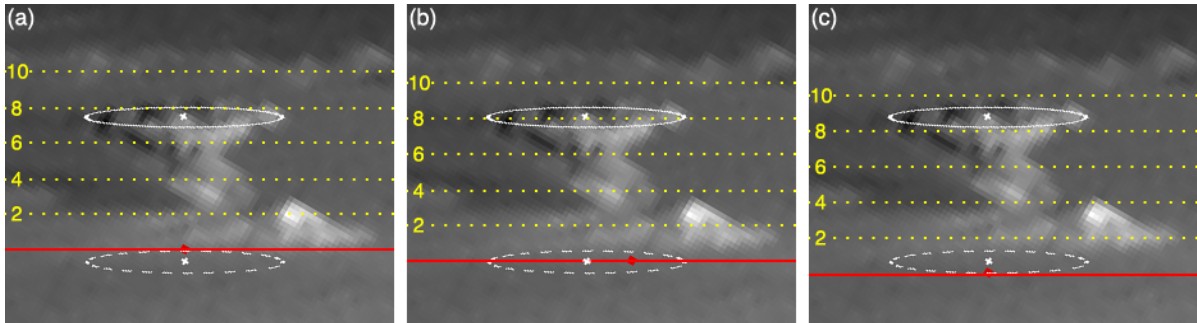

**Figure 3.** Effect of shifting the plane of isoheights, illustrated on the 19:10 UTC GOES-17 band 2 image of the 8 April 2020 eruption of Sheveluch. In all three panels the solid white line is the circular fit to the umbrella cloud from Fig. 2d and the dashed white line is the surface projection of that fit. The baseline (zero line) of the isoheights (red diamond and line) is fixed to **(a)** the westernmost point of the surface-projected fit, **(b)** the volcano base (default) or equivalently the center of the surface-projected fit, and **(c)** the easternmost point of the surface-projected fit. Shifting the plane of the isoheights in panels **(a)** and **(c)** back and forth compared to panel **(b)** roughly accounts for the westward/eastward expansion of the far-side/near-side of the umbrella cloud. With such base adjustments, the isoheights consistently indicate an elevation of ~8 km for the middle as well as the edges of the umbrella cloud.


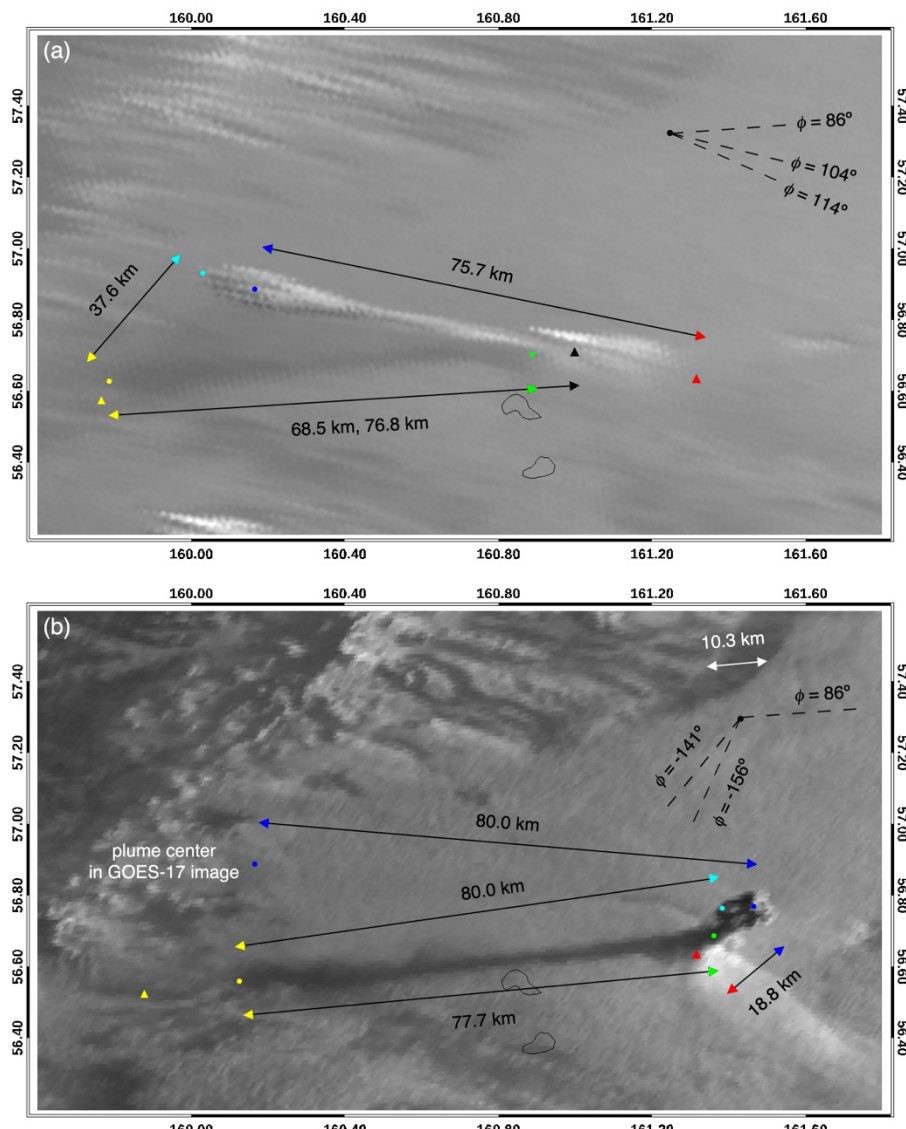

**Figure 4.** Equirectangular map projection of the 19:10 UTC **(a)** GOES-17 band 2 and **(b)** Himawari-8 band 3 image of the 8 April 2020 eruption of Sheveluch. The red and black triangles mark the volcano base and the vent, respectively. The green dot indicates the start point of the visible shadow, which for the Himawari-8 image is nearly the same as the vent location (thus, the black triangle is not shown in panel **(b)**), while the yellow dot is the shadow terminus on the cloud layer. The yellow triangle is the estimated position of the shadow terminus had it been cast directly on the surface without the intervening cloud layer. The blue and cyan dots respectively correspond to the center and the westernmost edge of the umbrella cloud; the GOES-17 image location of the plume center is shown in the Himawari-8 image too. Double-headed arrows indicate the ellipsoid-projected distance and direction between the points matching the color of the arrowheads. The white arrow marks the shadow of the low-level cloud layer on which the eruption column casts its shadow. The true and map distorted view azimuths at the volcano base ($\phi = 114°$ and $\phi = 104°$ for GOES-17, and $\phi = -156°$ and $\phi = -141°$ for Himawari-8) as well as the mapped solar azimuth ($\phi = 86°$, little distortion) are also indicated.





760

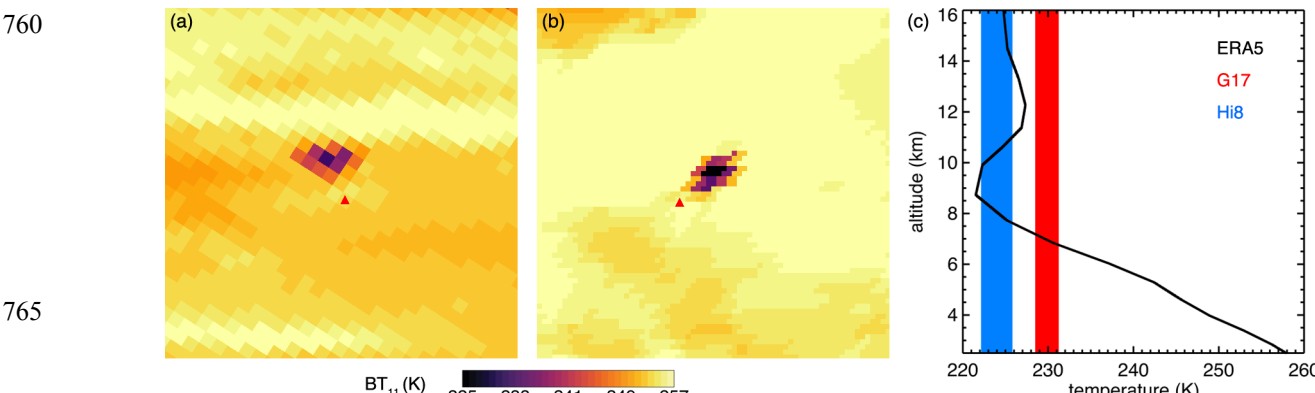

765

**Figure 5.** Band 14, 11.2-µm brightness temperature ($BT_{11}$) image of the 8 April 2020 eruption of Sheveluch at 19:10 UTC from **(a)** GOES-17 fixed grid data and **(b)** Himawari-8 CEReS V20190123 gridded and resampled data (0.02° or ~2 km
770 resolution). The red triangle marks the volcano base. In panel **(c)** the black line is the ERA5 temperature profile above the vent (2.5 km altitude). The red and blue shaded areas respectively indicate the range of the GOES-17 and Himawari-8 minimum $BT_{11}$ between 19:10 UTC (warmest dark pixel) and 19:50 UTC (coldest dark pixel).



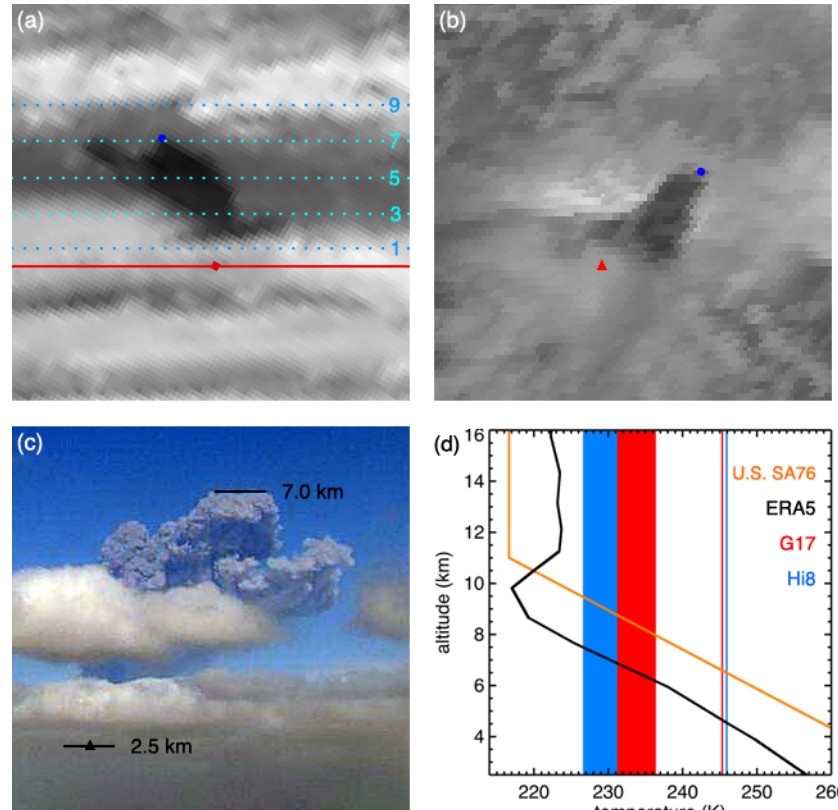

**Figure 6. (a)** GOES-17 band 2 fixed grid image (8x magnification), **(b)** Himawari-8 band 3 CEReS V20190123 gridded and resampled image (0.005° or ~0.5 km resolution, 3x magnification), and **(c)** Klyuchi webcam image of the 10 April 2019 eruption of Sheveluch at 03:00 UTC. The red diamond/triangle marks the volcano base and the blue dots indicate the same plume top element in the satellite images. In panel **(a)**, the baseline (solid red) and the odd number isoheights (in km, dotted cyan) are also drawn. In panel **(c)**, the black triangle denotes the vent. Panel **(d)** shows the ERA5 temperature profile above the vent (black line), the U.S. Standard Atmosphere 1976 temperature profile (orange line), the GOES-17 and Himawari-8 minimum $BT_{11}$ at 03:00 UTC (red and blue vertical lines), and the range of the minimum $BT_{11}$ between 03:10–04:00 UTC (red and blue shaded areas).





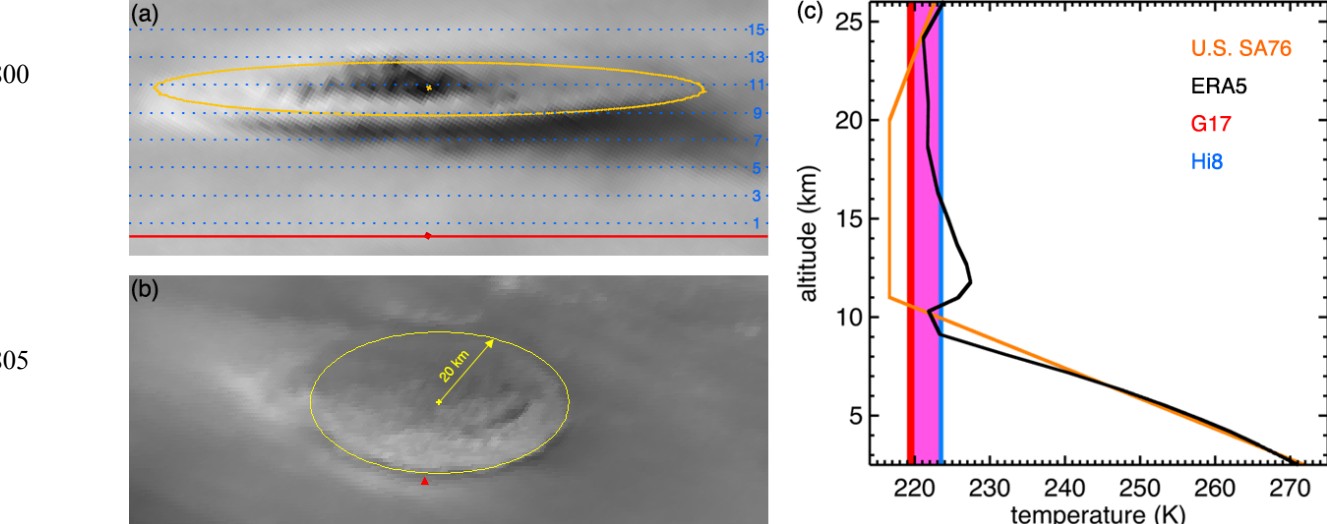

**Figure 7. (a)** GOES-17 band 2 fixed grid image (8x magnification) and **(b)** Himawari-8 band 3 CEReS V20190123 image of the 29 August 2019 eruption of Sheveluch at 03:10 UTC. The red diamond/triangle marks the volcano base and the solid yellow line is a circle of 20 km radius fitted to the umbrella cloud. In panel **(a)**, the baseline (solid red) and the odd number isoheights (in km, dotted blue) are also drawn. Panel **(c)** shows the ERA5 temperature profile (black line), the U.S. Standard Atmosphere 1976 temperature profile (orange line), and the range of the GOES-17 and Himawari-8 minimum $BT_{11}$ between 02:00–04:00 UTC (red and blue shaded areas, overlap in magenta).


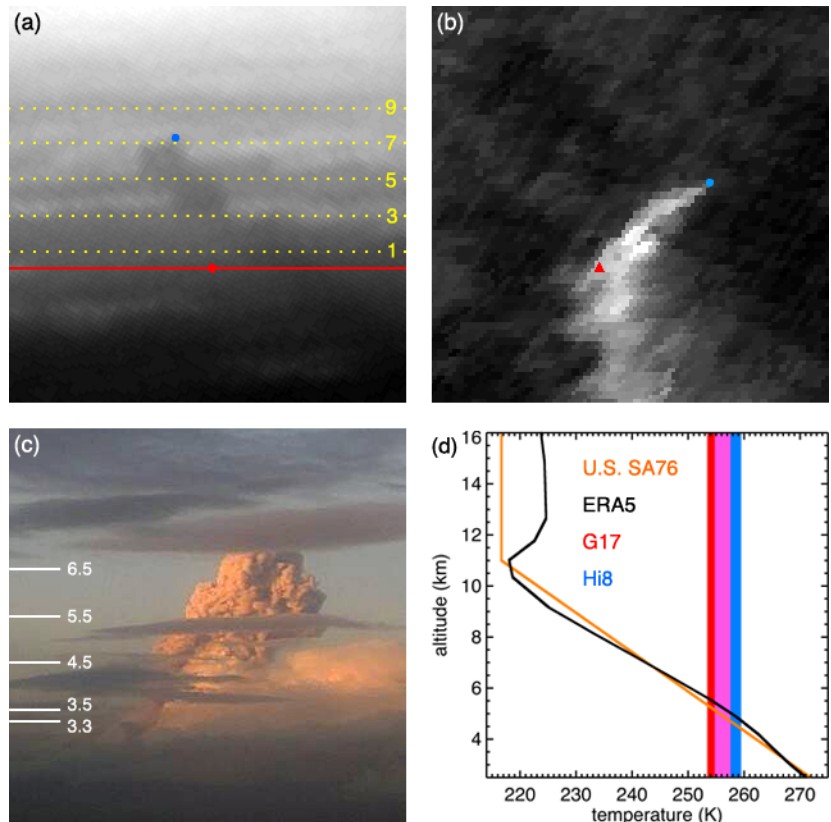

**Figure 8. (a)** GOES-17 band 2 fixed grid image (8x magnification, contrast enhanced), **(b)** Himawari-8 band 3 CEReS V20190123 image (contrast enhanced), and **(c)** Klyuchi webcam image of the 30 August 2019 eruption of Sheveluch at 08:00 UTC. The red diamond/triangle marks the volcano base and the blue dots indicate the same plume top element in the satellite images. In panel **(a)**, the baseline (solid red) and the odd number isoheights (in km, dotted yellow) are also drawn. In panel **(c)**, the white markings indicate height in km. Panel **(d)** shows the ERA5 temperature profile (black line), the U.S. Standard Atmosphere 1976 temperature profile (orange line), and the range of the GOES-17 and Himawari-8 minimum $BT_{11}$ between 08:00–09:00 UTC (red and blue shaded areas, overlap in magenta).



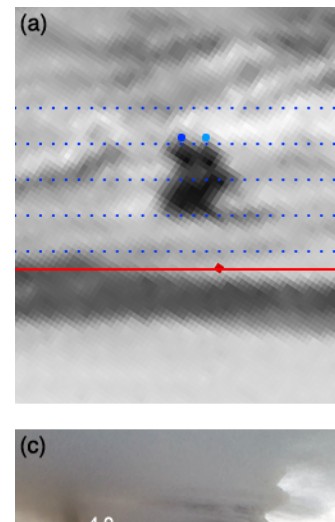

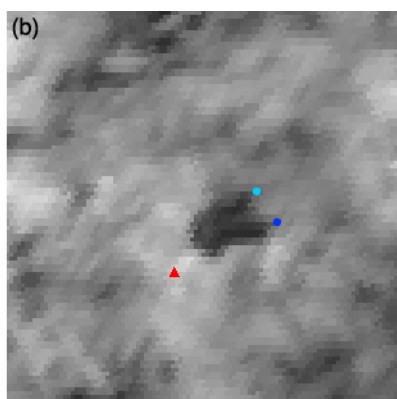

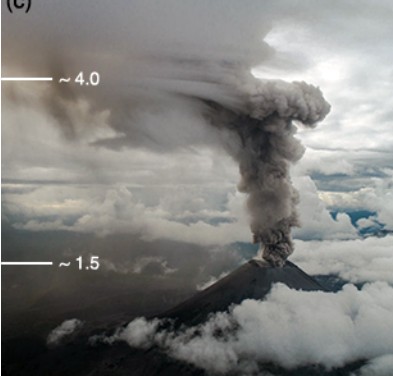

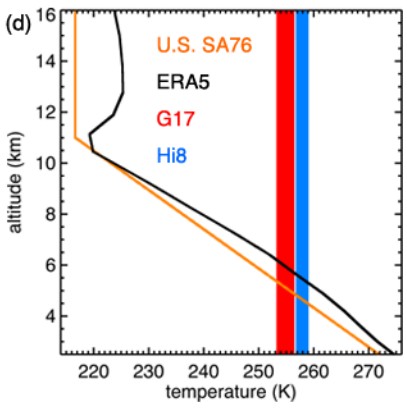

**Figure 9.** The 14 August 2019 eruption of Karymsky: **(a)** GOES-17 band 2 fixed grid image (8x magnification) at 04:30 UTC, **(b)** Himawari-8 band 3 CEReS V20190123 image at 04:30 UTC, and **(c)** quadcopter image at 04:45 UTC. The red diamond/triangle marks the volcano base, while the light and dark blue dots indicate two identifiable plume top elements in the satellite images. In panel **(a)**, the baseline (solid red) and the odd number isoheights (in km, dotted blue) are also drawn. In panel **(c)**, the white markings indicate height in km. Panel **(d)** shows the ERA5 temperature profile (black line), the U.S. Standard Atmosphere 1976 temperature profile (orange line), and the range of the GOES-17 and Himawari-8 minimum $BT_{11}$ between 04:30–05:30 UTC (red and blue shaded areas).





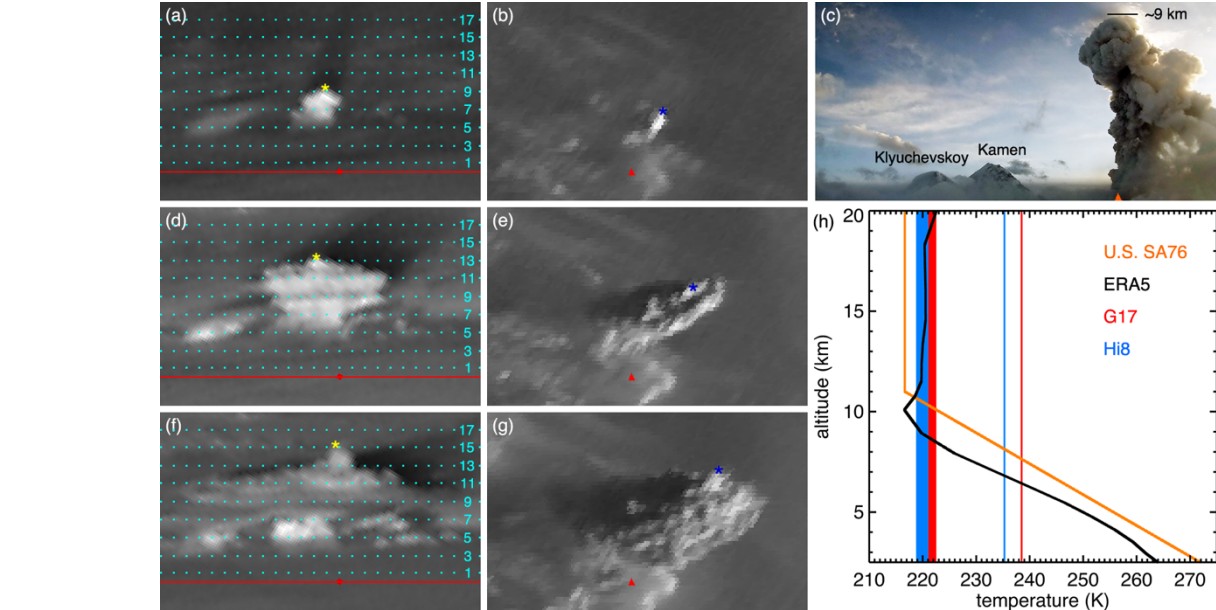

**Figure 10.** The 21 October 2020 eruption of Bezymianny: GOES-17 band 2 fixed grid image (8x magnification) at **(a)** 20:30 UTC, **(d)** 20:40 UTC, and **(f)** 20:50 UTC, the corresponding Himawari-8 band 3 CEReS V20190123 image at **(b)** 20:30 UTC, **(e)** 20:40 UTC, and **(g)** 20:50 UTC, and **(c)** the webcam image from the Kirishev seismic station at 20:29 UTC. The red diamond/triangle marks the volcano base, while the yellow and blue asterisks indicate the plume top in the satellite images. In panels **(a), (d),** and **(f)**, the baseline (solid red) and the odd number isoheights (in km, dotted cyan) are also drawn. Panel **(h)** shows the ERA5 temperature profile (black line), the U.S. Standard Atmosphere 1976 temperature profile (orange line), the GOES-17 and Himawari-8 minimum $BT_{11}$ at 20:30 UTC (red and blue vertical lines), and the range of the minimum $BT_{11}$ between 20:40–22:30 UTC (red and blue shaded areas).



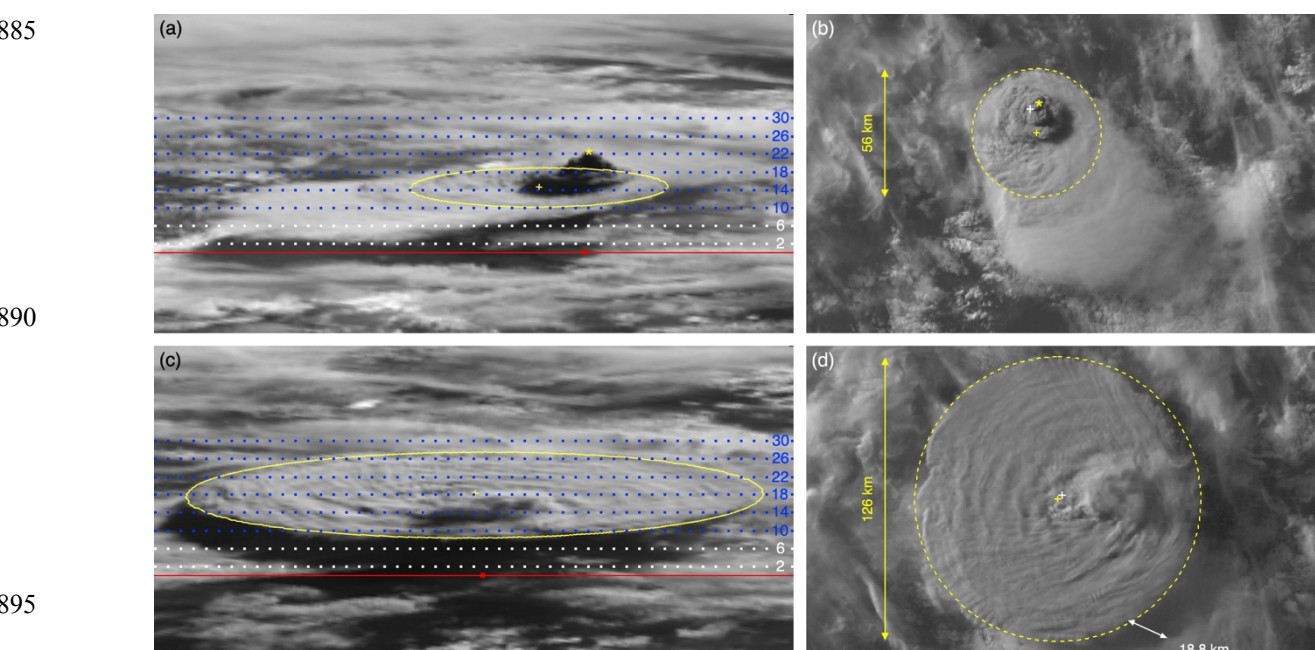

**Figure 11.** The 26 June 2019 eruption of Ulawun: GOES-17 band 2 fixed grid image (4x magnification) at **(a)** 06:10 UTC and **(c)** 07:00 UTC and the corresponding Himawari-8 band 3 CEReS V20190123 image at **(b)** 06:10 UTC and **(d)** 07:00 UTC. The red square (GOES-17) and white plus sign (Himawari-8) mark the volcano base, while the solid/dashed yellow line is a circle fitted to the umbrella cloud with a radius of 28 km at 06:10 UTC and 63 km at 07:00 UTC (the yellow plus sign is the circle center). The yellow asterisk indicates the overshooting top in the 06:10 UTC images. In panel **(d)**, the white arrow depicts the length and direction of the shadow the upper umbrella cloud casts on the lower umbrella cloud. In panels **(a)** and **(c)**, the baseline (solid red) and selected isoheights (in km, dotted white/blue) are also drawn.





910

915

**Figure 12.** The ERA5 temperature profile (black line), a tropical standard atmosphere temperature profile (green line), and the range of the GOES-17 and Himawari-8 minimum $BT_{11}$ between 05:50–07:50 UTC (red and blue shaded areas, overlap in magenta) for the 26 June 2019 eruption of Ulawun.





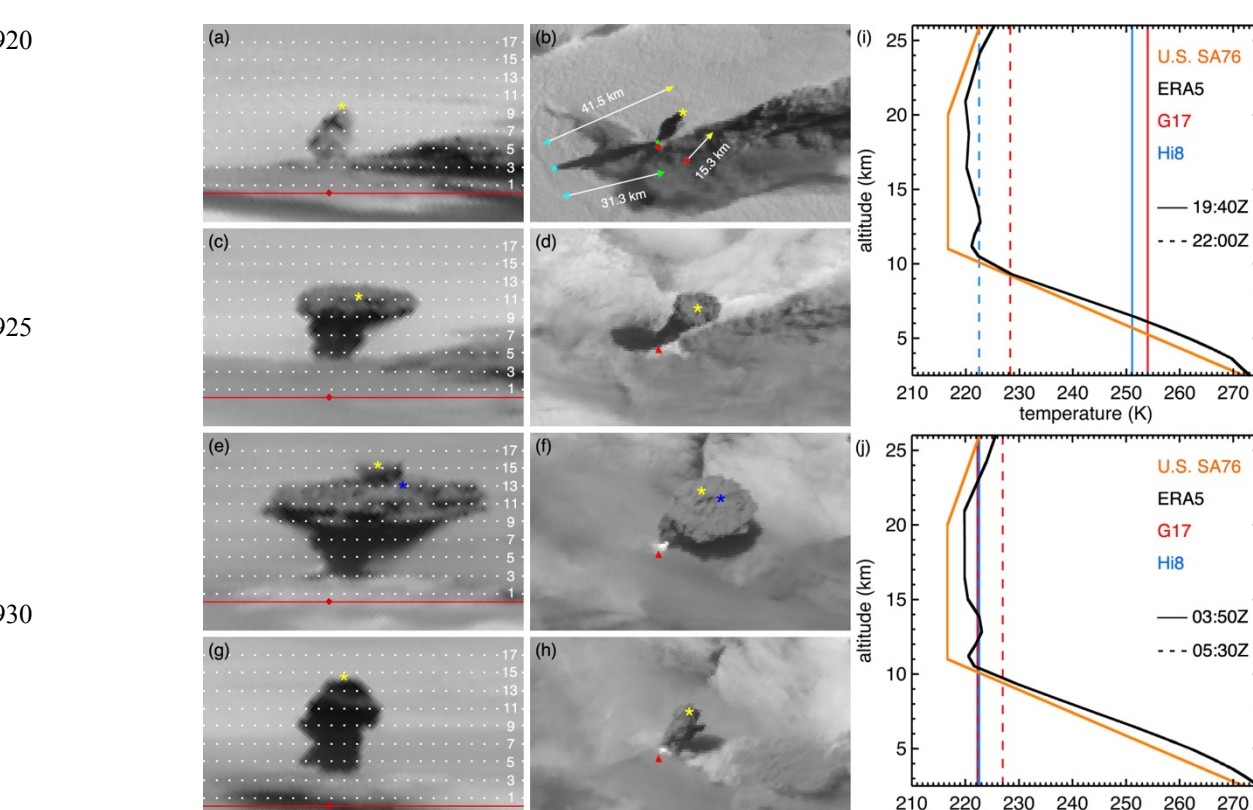

**Figure 13.** The 21-22 June 2019 eruption of Raikoke: GOES-17 band 2 fixed grid image (8x magnification) at **(a)** 19:40 UTC on 21 June, **(c)** 22:00 UTC on 21 June, **(e)** 03:50 UTC on 22 June, **(g)** 05:30 UTC on 22 June and the corresponding Himawari-8 band 3 CEReS V20190123 images in panels **(b)**, **(d)**, **(f)**, and **(h)**. The red diamond/triangle marks the volcano base, while the yellow asterisks indicate the same plume top feature in the satellite images. In panels **(e)** and **(f)**, the yellow and blue asterisks denote the overshooting top and a point on the umbrella cloud, respectively. In panel **(b)**, the green dot is the start point of the shadow, the cyan dot is the shadow terminus on the marine Sc layer, and the double-headed arrows indicate the ellipsoid-projected distance and direction between the points matching the color of the arrowheads. The baseline (solid red) and the odd number isoheights (in km, dotted white) are also drawn in the GOES-17 images. Panels **(i)** and **(j)** show the ERA5 temperature profile (black line) at 21:00 UTC on 21 June and 05:00 UTC on 22 June, the U.S. Standard Atmosphere 1976 temperature profile (orange line), and the GOES-17 and Himawari-8 minimum $BT_{11}$ (red and blue vertical lines) corresponding to the satellite images.



950

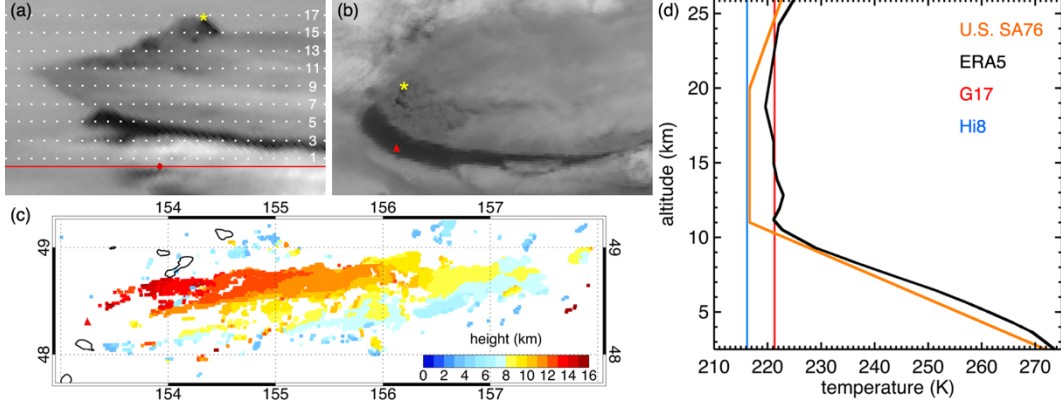

955 **Figure 14.** The Raikoke plume on 22 June 2019 at 01:20 UTC: **(a)** GOES-17 band 2 fixed grid image (8x magnification), **(b)** Himawari-8 band 3 CEReS V20190123 image, and **(c)** MODIS *Terra*–Himawari-8 3D Winds stereo heights. The red diamond/triangle marks the volcano base, while the yellow asterisks indicate the same overshooting top feature in the satellite images. The baseline (solid red) and the odd number isoheights (in km, dotted white) are also drawn in panel **(a)**. Panel **(d)** shows the ERA5 temperature profile (black line) at 01:00 UTC on 22 June, the U.S. Standard Atmosphere 1976 960 temperature profile (orange line), and the GOES-17 and Himawari-8 minimum $BT_{11}$ (red and blue vertical lines) corresponding to the satellite images.





**Table 1.** Case study plume height estimates, in kilometer and rounded to the nearest hectometer, derived from GOES-17 (G17) and Himawari-8 (Hi8) data. The retrieval methods are described in Part 1. The three traditional geometric methods are as follows. Method 1 (M1): sensor-projected length, method 2 (M2): true (stick) shadow length, and method 3 (M3): edge shadow length; method 3 can also be applied in 'stereo mode' using the parallax between the GOES-17 and Himawari-8 image locations of a given plume feature. Height is also estimated by matching the minimum 11-μm brightness temperature ($BT_{11}$) to the ERA5 temperature profile; if the reanalysis profile yields no solution due to plume undercooling (X), the height derived from the U.S. Standard Atmosphere 1976 temperature profile is given instead in parenthesis. VolSatView is the temperature method used by KVERT. The last column is the new geometric side view estimate, obtained from GOES-17 fixed grid imagery.

| | M1 G17 | M1 Hi8 | M2 G17 | M2 Hi8 | M3 G17 | M3 Hi8 | M3 stereo G17–Hi8 | $BT_{11}$ G17 | $BT_{11}$ Hi8 | VolSatView Hi8 | Side view G17 |
|---|---|---|---|---|---|---|---|---|---|---|---|
| Sheveluch 2020-04-08, 19:10Z | 8.2 | 7.9 | 8.0[a] | 8.3[a] | 7.6[a] | 7.8[a] | 8.0 | 6.8 | 7.6, 11.0, 14.0 | 9.6 | 8.0 |
| Sheveluch 2019-04-10, 03:00Z | 7.1 | 6.9 | | | | | 7.0 | 4.6 | 4.5 | 6.5 | 7.2 |
| Sheveluch 2019-08-29, 03:10Z | | | | | | | | X / (10.6) | X / (10.4) | 10.5 | 10.8 |
| Sheveluch 2019-08-30, 08:00Z | 7.3 | 7.0 | | | | | 6.8 | 5.0 | 4.7 | 6.5 | 7.2 |
| Karymsky 2019-08-14, 04:30Z | 6.8 | 7.0 | | | | | 6.9, 7.3 | 5.9 | 5.5 | 4.7 | 7.1 |
| Bezymianny 2020-10-21, 20:30Z | 9.4 | 8.7 | | | | | 9.1 | 6.4 | 6.8 | 7.7 | 9.3 |
| Bezymianny 2020-10-21, 20:40Z | 13.5 | 12.4 | | | | | 13.1 | 8.5, 19.7 | 8.9, 12.6 | 10.6 | 13.2 |
| Bezymianny 2020-10-21, 20:50Z overshooting top | 15.7 | 15.5 | | | | | 15.2 | 8.6, 19.5 | 9.1, 11.2 | 10.7 | 15.3 |
| Ulawun 2019-06-26, 06:10Z overshooting top | 21.5 | | | | | | 21.3 | 16.1, 17.6 | X / (X) | | 22.3 |





| | | | | | | | | | |
|---|---|---|---|---|---|---|---|---|---|
| Ulawun 2019-06-26, 07:00Z umbrella cloud | | | | | | 16.4, 17.1 | 16.5, 16.9 | | 18.3 |
| Raikoke 2019-06-21, 19:40Z | 9.7 | 10.0 | 9.6[b] | 9.8[b] | 10.0 | 6.1 | 6.5 | 8.8 | 9.9 |
| Raikoke 2019-06-21, 22:00Z | | | | | 11.6 | 9.4 | 10.7, 12.2, 13.7, 23.7 | 10.0 | 11.3 |
| Raikoke 2019-06-22, 01:20Z overshooting top | | | | | 16.1 | 11.3, 14.7, 22.5 | X / (X) | 11.1 | 16.5 |
| Raikoke 2019-06-22, 03:50Z overshooting top | | | | | 15.8 | 10.4, 12.2, 14.1, 23.1 | 10.4, 12.5, 13.8, 23.4 | 10.1 | 15.6 |
| Raikoke 2019-06-22, 03:50Z umbrella cloud | | | | | 13.7 | 10.4, 12.2, 14.1, 23.1 | 10.4, 12.5, 13.8, 23.4 | 10.1 | 13.0–14.0 |
| Raikoke 2019-06-22, 05:30Z | | | | | 14.6 | 9.7 | 10.4, 12.4 13.4, 22.8 | 10.2 | 14.5 |

[a]Low-level CTH estimate of 1.2 km added
[b]Low-level CTH estimate of 0.7 km added