# Peer review of "Geometric estimation of volcanic eruption column height from GOES-R near-limb imagery – Part 2: Case studies"

_Atmospheric Chemistry and Physics, 2021_

## Author Response (AR1)

**Part 2**

We thank the Referees for their constructive comments. In the following, we give a point-by-point response to each of the issues raised.

**Referee #1**

*L 46: Himawari-8 dark pixel BT11 with ERA5 profiles is used too. Right?*

Yes. We have corrected the sentence accordingly.

*Table 1: Why didn't you also put the webcam/quadcopter heigths results in table 1?*

We have added an extra column to Table 1, showing the webcam/quadcopter heights.

*Figure 14c and S3: the red triangle indicating the volcano is not clearly visible, is it possible to enlarge it or/and make it with a different color?*

We have enlarged the triangle and also added the letter 'R' next to it to more clearly indicate the location of Raikoke in these figures.

**Referee #2**

*Fig1: make larger symbols for volcanoes, it is difficult to see them.*

We have increased the size of the triangles and of the capital letters indicating the volcanoes.

*Fig3 and the following figs: red diamond is almost invisible on a red line, change colour of the line.*

We have changed the color of the line and occasionally that of the diamond too in all relevant figures, including Figs. 8 and 10 in Part 1 and supplementary Figs. S1 and S2. The baseline and the volcano are now marked in different colors.

*515: "3D Winds" method you apparently describe in the part 1 is not totally novel as suggested, see the link below, the authors used a triplet of two sequential SEVIRI and a MODIS image to consider the influence of wind on the height estimation.*
*https://acp.copernicus.org/articles/13/2589/2013/*
*The same methodology has been applied also on a combination of images from geostationary orbits:*
*https://www.mdpi.com/2072-4292/12/3/371*

We have deleted the word 'novel' and added a sentence to this paragraph, noting that a similar stereo method has previously been developed for Meteosat imagery by Zakšek et al. (2013) and Dehnavi et al. (2020).